# The combined action of the intracellular regions regulates FGFR2 kinase activity

Chi-Chuan Lin [1], Lukasz Wieteska[1], Guillaume Poncet-Montange[2], Kin Man Suen [1], Stefan T. Arold[3,4], Zamal Ahmed [5] & John E. Ladbury [1✉]

Receptor tyrosine kinases (RTKs) are typically activated through a precise sequence of intracellular phosphorylation events starting with a tyrosine residue on the activation loop (A-loop) of the kinase domain (KD). From this point the mono-phosphorylated enzyme is active, but subject to stringent regulatory mechanisms which can vary dramatically across the different RTKs. In the absence of extracellular stimulation, fibroblast growth factor receptor 2 (FGFR2) exists in the mono-phosphorylated state in which catalytic activity is regulated to allow rapid response upon ligand binding, whilst restricting ligand-independent activation. Failure of this regulation is responsible for pathologic outcomes including cancer. Here we reveal the molecular mechanistic detail of KD control based on combinatorial interactions of the juxtamembrane (JM) and the C-terminal tail (CT) regions of the receptor. JM stabilizes the asymmetric dimeric KD required for substrate phosphorylation, whilst CT binding opposes dimerization, and down-regulates activity. Direct binding between JM and CT delays the recruitment of downstream effector proteins adding a further control step as the receptor proceeds to full activation. Our findings underscore the diversity in mechanisms of RTK oligomerisation and activation.

[1] School of Molecular and Cellular Biology, and Astbury Centre for Structural Molecular Biology, University of Leeds, Leeds LS2 9JT, UK. [2] Center for the Development of Therapeutics, Broad Institute of MIT & Harvard, Cambridge, MA 02142, USA. [3] King Abdullah University of Science and Technology, Computational Bioscience Research Center, Division of Biological and Environmental Sciences and Engineering, Thuwal 23955-6900, Saudi Arabia. [4] Centre de Biochimie Structurale, CNRS, INSERM, Université de Montpellier, 34090 Montpellier, France. [5] Department of Molecular and Cellular Oncology, University of Texas, MD Anderson Cancer Center, Houston, TX 77030, USA. ✉email: j.e.ladbury@leeds.ac.uk

The tight regulation of the enzymatic activity of receptor tyrosine kinases (RTKs) is a fundamental precept of numerous cellular outcomes. Mutations which perturb this regulation can have devastating effects on signaling, leading to multiple pathologies including cancer, developmental abnormalities and metabolic disorders. Activity of the majority of RTKs investigated to date has been shown to be maximally up-regulated through asymmetric kinase domain (KD) dimerization, and alternating trans-autophosphorylation of tyrosines on each protomer, providing sites for effector protein recruitment[1–4]. However, additional control is exerted by intracellular amino acid sequences peripheral to KD, both within the juxtamembrane (JM), and the C-terminal tail region (CT) of the receptor. The modus operandi of these regions varies across different receptors and can lead to both down- and up-regulation of kinase activity[5].

Structural studies of unphosphorylated kinases have shown that binding of JM to KD results in the inhibition of kinase activity of several RTKs (e.g., PDGFR;[6] Eph-family RTKs;[7] MuSK;[8,9] Flt3;[10] Kit[11,12]). One well characterized example is the ephrin receptor B2 (EphB2) in which the interaction between JM and KD down-regulates activity through stabilization of the inactive conformation and constraint of the activation loop (A-loop)[7]. In contrast, the full activity of epidermal growth factor receptor (EGFR) requires the presence of JM which links the asymmetric dimer via a 'latch' sequence[2,13]. The impact of CT on KD regulation has also been shown to be important in several RTKs. For instance, CT inhibits access of substrates to KD in the unphosphorylated Tie2 receptor[14]. CT also supresses the catalytic activity of EGFR through stabilization of an inactive symmetric dimer[1,15–18].

The importance of CT in controlling pathogenic signal transduction is underscored in the expression of the oncogenic fibroblast growth factor receptor 2 (FGFR2) K-samII gene[19]. There are three variants of K-samII which produce different length truncations of CT. Cells in which the truncated K-sam gene is amplified exhibit a growth advantage in gastric cancers. Furthermore, transfection of the K-samII C-terminal truncated gene in NIH3T3 cells results in ligand-independent transforming activity[20]. Expression of a truncated variant in T24 bladder cells leads to un-regulated proliferation[21]. Recent data have further revealed that the deletion of the FGFR2 CT through exon truncation provides a driver alteration in cancer but also increases sensitivity to a subset of kinase inhibitors[22]. Thus, the presence of CT prevents pathogenic proliferative signaling from FGFR2 through an imprecisely known molecular mechanism.

Activation of typical RTKs is mediated through a pre-defined, sequential order of phosphorylation on the KD and CT starting from the A-loop[23–25]. We have previously shown that in the absence of stimuli, a tyrosine residue in the A-loop is phosphorylated in FGFR2 (Y657 in FGFR2IIIb)[26,27]. Since phosphorylation of this residue renders the kinase active, and yet activity is impeded, the mono-phosphorylated state could be considered as an 'active intermediate' state, i.e., the catalytic ability of KD is initiated, but subject to control. This state does not appear to prevail in EGFR, which is the best-characterised RTK to date. Indeed, A-loop tyrosine phosphorylation is not a requirement for EGFR activation[2,5,28,29]. However, an increasing number of studies have shown that, in the absence of ligand stimulation, many RTKs are able to self-associate into signaling incompetent dimers and the extent of phosphorylation on the kinase domain is restricted to a single tyrosine within the A-loop[30–35]. Therefore, investigation of the regulation of the mono-phosphorylated state of FGFR2 provides a precedent for understanding regulation of other RTKs.

Under basal conditions FGFR2 CT, which includes the proline-rich sequence PCLPQYP, binds to the C-terminal SH3 domain of dimeric growth factor receptor binding protein (GRB2) holding two molecules of FGFR2 in a signaling incompetent heterotetramer[26,27,30]. On stimulation interactions between KDs stabilize the active structure and permit the sequential phosphorylation of CT and KD[23,36]. Stimulation also results in JM providing a site for the recruitment of the scaffold protein FRS2 (including residues V429 and T430; the VT motif[37]). Concomitantly, the active receptor also phosphorylates GRB2 resulting in its dissociation[26,27,30] leaving CT exposed for tyrosine phosphorylation and downstream effector protein interaction.

Our current knowledge of the regulatory roles of JM and CT in typical RTKs is primarily restricted to experiments based on the unphosphorylated KD with either JM or CT independently. The influence of JM, CT and the phosphorylation state of KD on regulation of RTKs is unclear. In particular concerning regulation of the mono-phosphorylated active intermediate state. FGFR2 provides a good example of a highly regulated RTK, particularly since the A-loop Y657-mono-phosphorylated state prevails under unliganded conditions. We have adopted a reductionist approach, including deconstruction of intracellular FGFR2, to show how JM and CT contribute to the control of FGFR2 activity. We reveal the intricate mechanism by which the interplay of JM and proline-rich sequences on CT enable the receptor to sustain A-loop phosphorylation under non-stimulated conditions and yet inhibit further catalytic activity. In this state FGFR2 is primed to respond rapidly to growth factor binding to produce the phosphorylated platform for recruitment of downstream effector proteins but is subjected to stringent controls. Since approximately half of all RTKs include proline-rich sequences within their CT regions, some elements of this fine-tuning of regulation by JM and CT are likely to be conserved across RTKs. Our study demonstrates that multiple previously elusive interactions between the KD and intracellular regions of the receptor provide manifold control capabilities to fine tune outputs, strongly supporting the great diversity in the regulation of RTK function.

## Results

To provide structure and clarity to the understanding of the intricate control mechanism herein investigated, we have deconstructed intracellular FGFR2 to the three principle regulatory regions: JM, KD and CT. This provides an understanding of how both JM and CT influence KD activation and dimerization. Through subsequent reconstruction of their combined effects we could reconstitute a complete picture of the regulatory mechanism as a whole. Armed with these data we tested our mechanistic hypotheses on signaling outputs such as substrate turnover.

**Activity of FGFR2 KD is enhanced in the presence of JM.** The impact of JM (residues 414-465) on kinase function was investigated through four dephosphorylated FGFR2 JM-KD constructs with progressively increasing truncations of JM (schematic Fig. 1a). Truncation of JM dramatically reduced phosphorylation (Fig. 1a). However, deletion of the entire JM resulted in some phosphorylation of the KD product, as would be expected for an unencumbered kinase enzyme in solution. The influence of JM on phosphorylation was measured in HEK293T cells over-expressing full length FGFR2IIIb (C1 isoform, FGFR2$^{C1}$) including short, intermittent JM fragment deletions (schematic Fig. 1b). Immunoblotting for the A-loop phosphorylated tyrosines (pY657/pY658) revealed that both basal and FGF7-stimulated phosphorylation of FGFR2$^{C1}$ is significantly reduced in all JM deletion variants confirming the importance of intact JM (Fig. 1b).

To understand whether progressive sequential phosphorylation of KD affected JM binding we measured the affinities of the interactions between the MBP-JM and a series of six Y to F

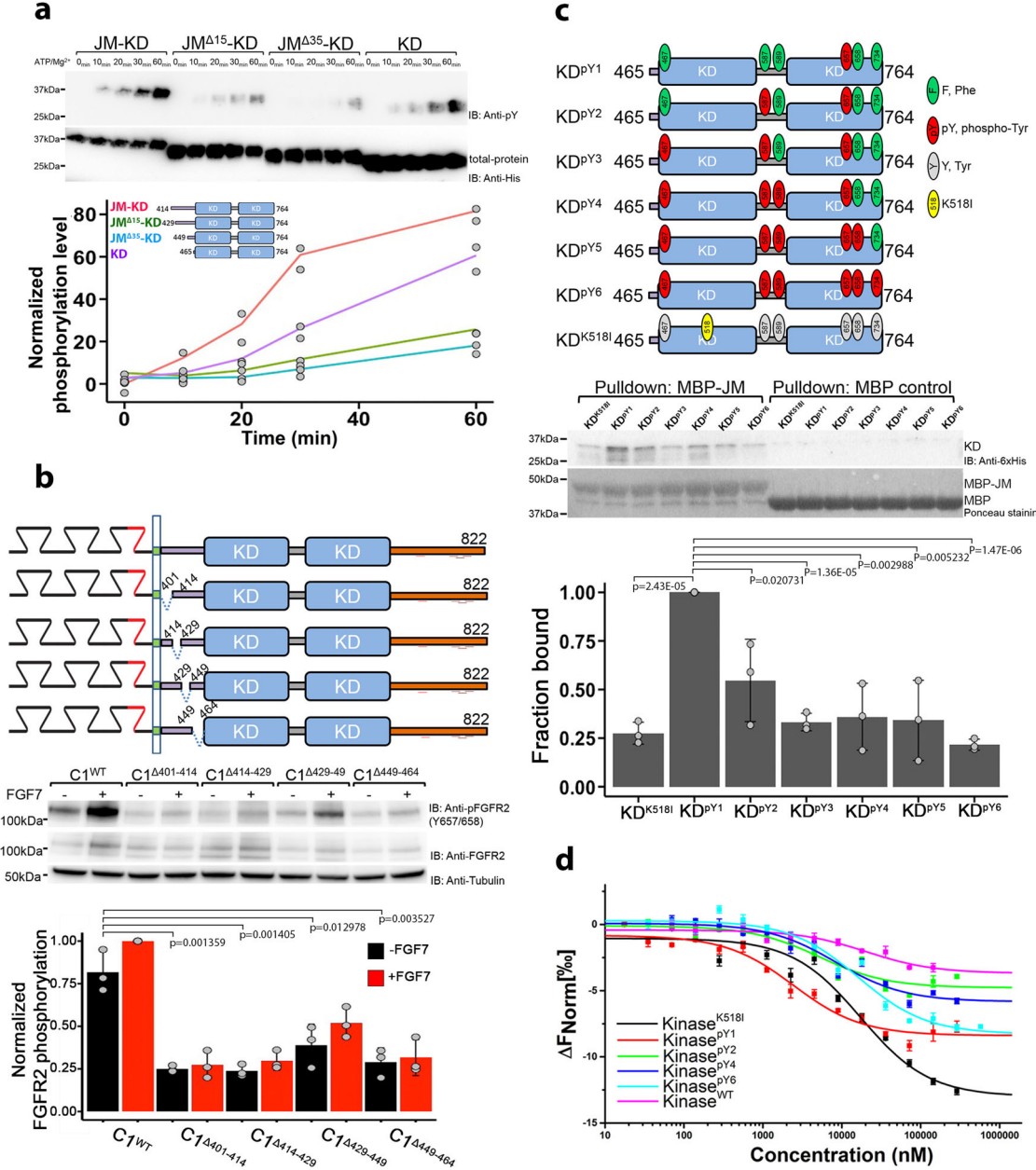

**Fig. 1 The presence of JM enhances activity of KD. a** In vitro kinase assay using progressive JM-deletions in JM-KD (residues 414–764, red; 429–764, green; 449–764, blue; and 465–764, purple). 100 nM of each protein was used for the in vitro phosphorylation assay. Phosphorylation levels were determined using a general pY99 antibody. His-tag antibody was used for total protein control. Densitometric line graph represents data from 2 independent experiments. **b** Intermittent deletions in JM down-regulate FGFR2 activity. HEK293T cells were transfected with FGFR2 with JM deletions (C1, C1$^{\Delta401-414}$, C1$^{\Delta414-429}$, C1$^{\Delta429-449}$, and C1$^{\Delta449-464}$). Cells were serum starved overnight and left unstimulated or stimulated with 10 ng/ml FGF7 for 15 min. Cell lysates were blotted with indicated antibodies. Densitometric bar graph represents three independent experiments. The error bars are presented as the standard deviation. **c** Binding of JM to progressively phosphorylated KD. Six tyrosine residues on KD were mutated to mimic the sequential phosphorylation pattern of KD (KD$^{pY1}$ to KD$^{pY6}$; Supplementary Fig. 1b). MBP-JM was used to pulldown KDs (His-tagged). Densitometric bar graph represents three independent experiments. The error bars are presented as the standard deviation. **d** MST measurements of the binding affinity between JM and KD with different phosphorylation levels. JM was labeled with Atto 488 dye and serial dilutions of KD were titrated at 25 °C. The error bars represent standard error of the mean from three technical replications.

mutants that mimic the sequential phosphorylation pattern of KD (KD$^{pY1}$ to KD$^{pY6}$;[22–24] schematic Fig. 1c and Supplementary Fig. 1a). JM binds most strongly to the mono-phosphorylated KD$^{pY1}$ (i.e., the 'active intermediate' state) as shown by pulldown (Fig. 1c) and measured by microscale thermophoresis, MST ($K_{d,app} = 2.51 \pm 0.20 \mu M$; Supplementary Table 1 and Fig. 1d). Only weak binding is apparent with a non-phosphorylatable

catalytically inactive K518I mutant, KD$^{K518I}$. The affinity of JM for KD reduces with progressive phosphorylation.

To identify the precise region of JM that binds to KD$^{pY1}$, we generated five peptides corresponding to consecutive sequences of 15–16 amino acids from JM and measured their affinities to KD$^{pY1}$ (Supplementary Table 2 and Supplementary Fig. 1b). Only two peptides bound to KD$^{pY1}$, and these share the

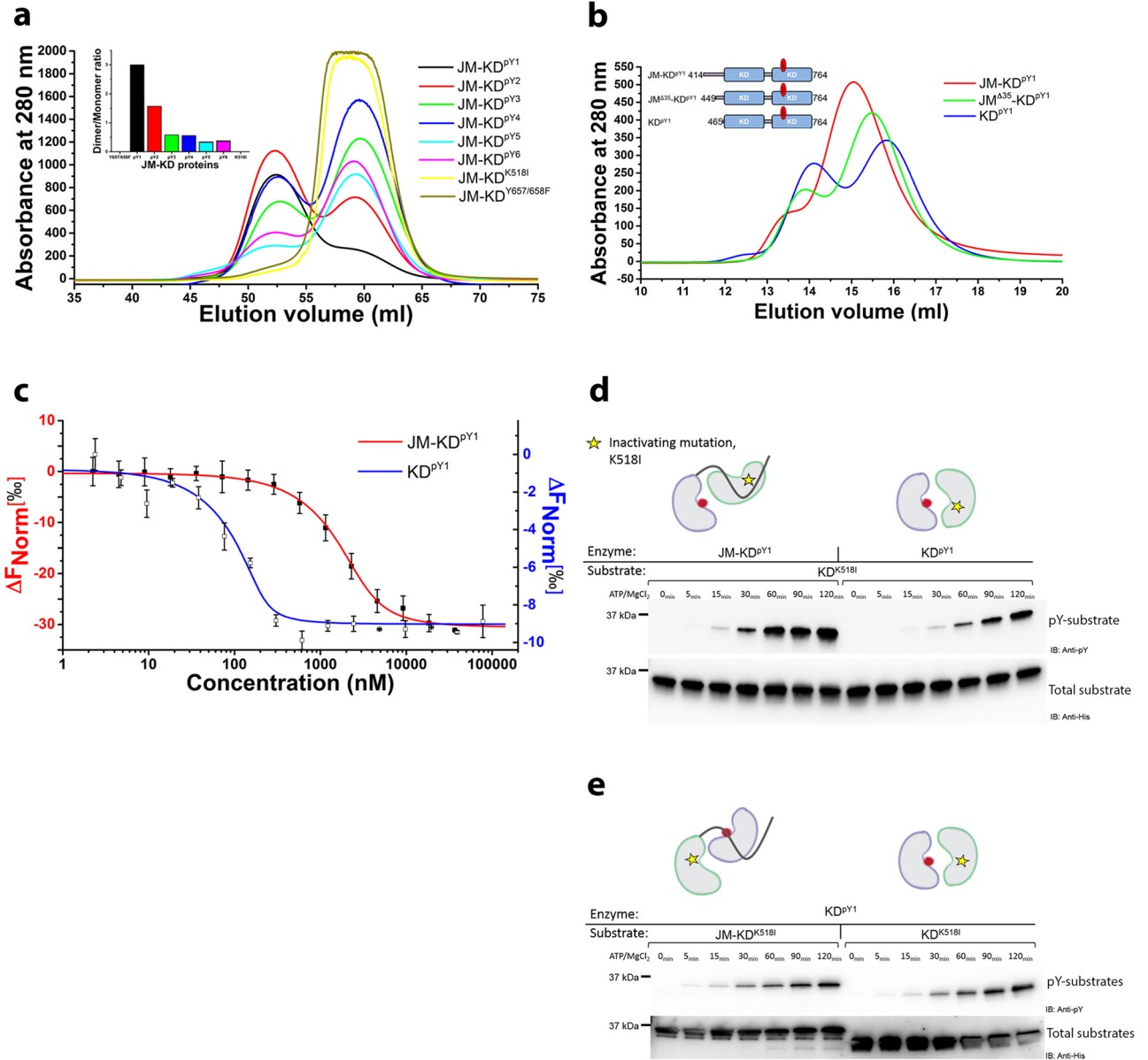

**Fig. 2 JM forms an intermolecular interaction in an asymmetric KD dimer. a** JM-KD construct with progressively increasing pY residues on KD were run on size exclusion chromatography (SEC; Superdex 75 26/100) at 80–100 μM (inset: dimer/monomer ratio of KD with different phosphorylation levels). **b** Dimerization of KD$^{pY1}$ is reduced in the presence of JM. The dimerization of JM-KD$^{pY1}$ constructs with JM deletions (JM-KD$^{pY1}$; JM$^{Δ35}$-KD$^{pY1}$; and KD$^{pY1}$) was determined using 10 μM injected on a size exclusion column (Superdex 75 10/30). **c** The 'apparent' dimerization K$_d$ of JM-KD$^{pY1}$ (red) and KD$^{pY1}$ (blue) determined using MST. JM-KD$^{pY1}$ and KD$^{pY1}$were labeled with Atto488 dye then titrated with unlabeled JM-KD$^{pY1}$ and KD$^{pY1}$. The error bars represent the standard deviation of 2 technical replications. **d** Under basal conditions the presence of JM in the enzyme-acting molecule (JM-KD$^{pY1}$) is required for the recruitment of substrate-acting molecule (JM-KD$^{K518I}$). This sustains the asymmetric dimer configuration required for the enhancement of transphosphorylation as the phosphorylation levels of substrates (left side: enzyme: JM-KD$^{pY1}$, right side: enzyme: KD$^{pY1}$) were examined using a phosphotyrosine antibody (pY99). An anti-6xHis tag antibody was used to probe total proteins as the loading control. **e** The JM from JM-KD$^{K518I}$ cannot recruit KD$^{pY1}$ to form an active dimer. Phosphorylation of substrate is at the same level as the independent monomers. Together Fig. 2d, e demonstrate a role of kinase activation under basal conditions where the JM interacts in trans, recruiting and phosphorylating a substrate molecule.

consensus sequence $^{414}$PAVHKLT$^{420}$ which is proximal to the N-terminal of JM.

**JM binding promotes asymmetric over symmetric KD$^{pY1}$ dimer formation**. Asymmetric dimerization is crucial for RTK enzymatic function. Having shown that JM binds to mono-phosphorylated KD and enhances substrate phosphorylation, we investigated the impact of JM on dimerization. A series of JM-KD polypeptides exhibiting progressively increasing phosphorylation states (schematic Supplementary Fig. 2a, and Supplementary

Fig. 2b) showed the highest population of dimer in the mono-phosphorylated state, JM-KD$^{pY1}$ by gel filtration (Fig. 2a; protein concentrations: 80–100 μM). In both the catalytically inactive mutants, JM-KD$^{K518I}$ and JM-KD$^{Y657/658F}$, dimerization was abrogated. The dependence of both JM binding and dimerization on the phosphorylation state of KD is consistent with JM acting as an intermolecular latch which is released with increasing pY burden.

To establish further how the JM affects the dimeric state of KD$^{pY1}$, we used three JM-KD$^{pY1}$ constructs with progressively

truncated JM (schematic Fig. 2b; polypeptide concentrations: 10 µM). Truncation of JM resulted in an increase in the population of dimers: isolated $KD^{pY1}$ shows the greatest population of dimer. It should be noted that the monomer/dimer ratio and the elution volumes in Fig. 2a, b are different due to the usage of different protein concentrations and different separation columns. We quantified the 'apparent' dimerization constant of the mono-phosphorylated KD in the absence ($KD^{pY1}$: $K_{d,app} = 112 \pm 9$ nM; Fig. 2c) and presence of JM ($JM\text{-}KD^{pY1}$: $K_{d,app} = 3.46 \pm 0.10$ µM by MST (Fig. 2c) and $K_{d,app} = 3.07$ µM by surface plasmon resonance, SPR (Supplementary Fig. 2c)). Thus, the presence of JM reduces the dimerization affinity by an order of magnitude, suggesting that $KD^{pY1}$ and $JM\text{-}KD^{pY1}$ dimers are conformationally different, and that the presence of JM reduces the ability of KD to tightly self-associate. In the absence of JM, $KD^{pY1}$ forms a symmetric dimer, and occludes access of substrate (as shown in X-ray crystal structure; Supplementary Fig. 2d, e and Supplementary Table 3). Thus, the presence of the JM-mediated latch appears to limit the KD self-association. We speculate that this provides a mechanism that enhances the ability of individual KDs to orientate with respect to one another to facilitate multi-site transphosphorylation.

To determine whether JM from the enzyme-like or substrate-like protomer forms the latch, we incubated catalytically inactive $KD^{K518I}$ (the substrate) with $JM\text{-}KD^{pY1}$ or $KD^{pY1}$ (the enzymes). In this case the presence of JM increased phosphorylation of $KD^{K518}$ (Fig. 2d). We then incubated $KD^{pY1}$ (the enzyme) with $JM\text{-}KD^{K518I}$ or $KD^{K518I}$ (the substrates) and observed no difference in phosphorylation of the two substrates (Fig. 2e). Thus, JM from the enzyme-like protomer binds to the substrate-like protomer in the asymmetric active dimer and increases the phosphorylation of the substrate-like molecule.

In summary, our data show that in the active intermediate state JM forms a latch from the enzyme-like protomer to the substrate-like protomer. In the absence of other regulatory interactions, this latch holds the active KDs such that they can asymmetrically interact with one another, whilst being prevented from higher affinity self-association. This observation is analogous to that seen in EGFR[2] however, our data further show that the latch becomes less stabilizing as FGFR2 KD phosphorylation progresses.

**Activity of KD is inhibited by CT**. Having defined the impact of JM on KD activity and dimerization we turn our attention to the regulatory function of CT. Initially, we measured the impact on receptor activity of N-terminally Flag-tagged, FGFR2IIIb C1, C2, C3 K-samII oncogenic isoforms. HEK293T cells transfected with $FGFR2^{C1}$; $FGFR2^{C2}$; $FGFR2^{C3}$ or $FGFR2^{C1\Delta34}$ ($FGFR2^{C1}$ with 34 amino acids deleted from the C-terminus that is identical in length to C2 but has the same sequence as C1, i.e., does not include two consecutive PXXP motifs found in C2: CT sequences shown in Fig. 3a) revealed that deletion of CT led to increased receptor phosphorylation and activation of effector proteins in the absence of growth factor stimulation (Fig. 3a). Deletion of the entire CT in $FGFR2^{C3}$ promotes downstream signaling in the ERK1/2 (MAPK) pathway without ligand stimulation. This could be due to the binding of the scaffold protein FRS2 which is known to bind to JM and mediate downstream effector protein recruitment to the activated C3 receptor[38] (in contrast to FGFR1 to which FRS2 is constitutively bound[39]). This result strongly suggests that the presence of CT controls FGFR2 kinase activity as well as the interaction of FRS2 with the receptor.

Consistent with the cell-based assay (Fig. 3a), the phosphorylation of the A-loop increases as CT is truncated in recombinant $KD\text{-}CT^{C1}$, $KD\text{-}CT^{C2}$, $KD\text{-}CT^{C3}$ or $KD\text{-}CT^{C1\Delta34}$ Fig. 3b). Again $KD\text{-}CT^{C3}$ is accompanied by increased phosphorylation. Thus,

$KD\text{-}CT^{C3}$, like $KD^{pY1}$ which appears in dynamic equilibrium between monomers and symmetric head-to-tail dimers (Fig. 2b and Supplemental Fig. 2d), behaves as a free enzyme (as in Fig. 1a). CT in $KD\text{-}CT^{C1\Delta34}$ also releases inhibition as seen in our cell-based assay. However, $KD\text{-}CT^{C2}$, which contains an identical length but includes the mutations producing PXXP motifs, restores the inhibitory capability. This points to an important role for CT proline-rich motifs.

**Interaction of CT with $KD^{pY1}$**. So far, our data indicate that when CT is present it inhibits the receptor. This could occur via two distinct mechanisms; 1) antagonistically blocking receptor activation through direct binding to KD, and/or 2) through binding of CT to JM and/or KD to inhibit formation of asymmetric dimer.

$CT^{C1}$ binds to $KD^{pY1}$ with moderate affinity ($K_d = 3.75 \pm 0.46$ µM; Fig. 3c, blue curve). NMR spectroscopy was used to probe the residues involved in the interaction of $CT^{C1}$. To this end, CSPs of $^{15}N$-labeled $CT^{C1}$ were measured on addition of $KD^{pY1}$ (Fig. 3d). Two distinct potential interacting regions of $CT^{C1}$ were observed; residues around 770 to 780 (e.g., Q775); and residues within the proline-rich motif in the C-terminus (D802 to Y813) (Fig. 3d). MST affinity measurements of peptide fragments of CT revealed that the tightest binding sequence was $^{801}PDPMPYEPCLPQYPH^{815}$ ($K_d = 25.9 \pm 5.4$ µM; Supplementary Table 4 and Supplementary Fig. 3a). Based on this, we concluded that the last 23 residues containing the proline-rich sequence are necessary for binding with the $KD^{pY1}$, and this facilitates concomitant engagement of the first 24 residues in the intact $CT^{C1}$. Our data contrasts with C-terminal binding and inhibition of KD activity in EGFR. The EGFR CT (~230 amino acids) is much longer than FGFR2 (~55) but the KD is regulated by the ~70 amino acid sequence proximal to the KD[40] whilst the additional ~160 residues (further from the KD) have alternative function.

An in vitro pulldown assay showed that both the GST-tagged $CT^{C1}$ and $CT^{C2}$ can interact with $KD^{pY1}$ (Fig. 3e). Binding was significantly reduced in the presence of point mutations of the proline residues to alanine except P801A and P814A. The mutation of both P803 and D802 have a large impact on binding. GST-$CT^{C2}$ bound to $KD^{pY1}$ whilst the first 24 amino acids (GST-$CT^{C1\Delta34}$) of C1 did not. The similarity of $CT^{C2}$ with the wild type $CT^{C1}$ was also apparent in the kinase phosphorylation data (Fig. 3e). Sequence alignment suggests that the interactions are strongest when the sequence includes a PXEPXXPXYP motif (where X is any residue) which occurs between residues 805 and 814 for $CT^{C1}$ and 776 and 785 for $CT^{C2}$.

Point mutations in $FGFR2^{C1}$ PDPMPXEPXXPXYP sequence confirmed the importance of this region for signaling in HEK293T cells (Supplementary Fig. 3b). Even in the absence of FGF7 the corruption of the proline-rich sequence has a dramatic affect in up-regulation of FGFR2 and its downstream ERK1/2 pathways. Inhibition of recombinant KD by peptides derived from residue 801–822 identified the shortest sequence of CT for KD down-regulation ($^{808}PCLPQYPH^{815}$, Supplementary Fig. 3c and Supplementary Table 4 for sequences). Expression of $^{808}PCLPQYPH^{815}$ in a human gastric carcinoma cell line, KATO-III, which endogenously express FGFR2IIIb C3 isoform also results in the inhibition of cell proliferation as shown in a MTT assay (Supplementary Fig. 3d).

**CT inhibits dimerization of JM-KD**. CT can block dimer formation as part of $KD^{pY1}$-CT (Fig. 4a). Independent binding of CT also inhibits dimerization of the extended JM-KD construct, as demonstrated by steady-state fluorescence resonance

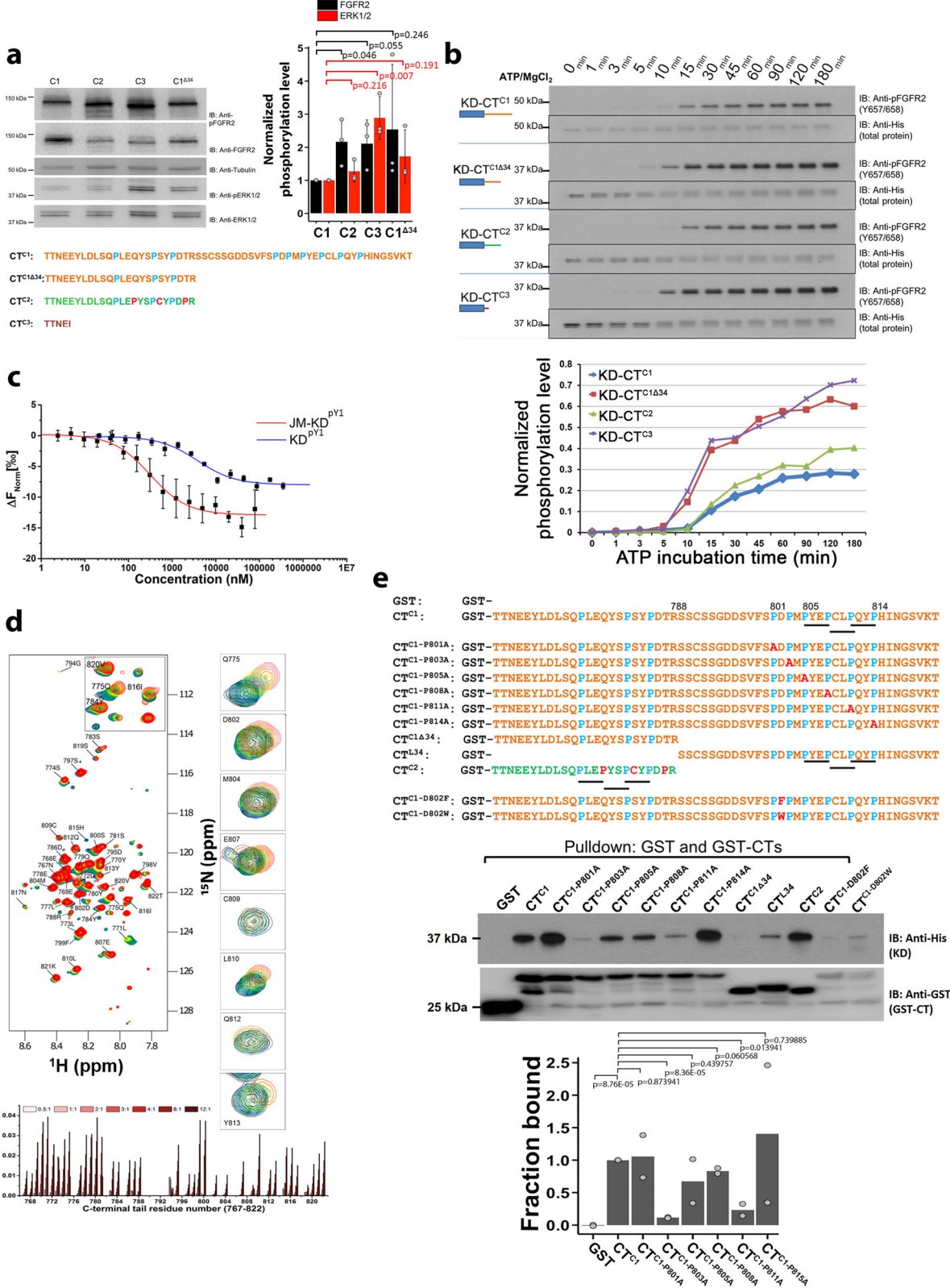

energy transfer (FRET) measurement (Supplementary Fig. 4a). GST-CT$^{C1}$ was able to pull down JM-KD$^{pY1}$ (Fig. 4b), and CT$^{C1}$ formed a high affinity complex with JM-KD$^{pY1}$ (MST, $K_d = 304 \pm 44$ nM; SPR, $K_d = 160 \pm 20$ nM; Supplementary Table 5, Fig. 3c (MST, red curve), Fig. 4b (SPR) and Supplementary Fig. 4b respectively). Knowing that CT disrupts the dimer formation, it is safe to assume that CT binds to the monomeric JM-KD$^{pY1}$ protomer. Changing the phosphorylation state of KD$^{pY1}$ (unphosphorylated or multi-phosphorylated), reduced the affinity of CT (Supplementary Fig. 4b).

To confirm the impact of the CT on dimer formation in cells we used fluorescence lifetime imaging microscopy (FLIM) in serum-starved HEK293T cells co-expressing both GFP- and RFP-tagged FGFR2$^{C1}$, FGFR2$^{C2}$ and FGFR2$^{C3}$. Compared to the

**Fig. 3 Proline-rich motifs interact and downregulate kinase activity. a** Immunoblotting analysis of signaling activity of FGFR2IIIb isoforms. FGFR2[C1]; FGFR2[C1Δ34]; FGFR2[C2] and FGFR2[C3] were transfected into HEK293T cells starved overnight. The levels of receptor phosphorylation and downstream ERK phosphorylation on each isoform were probed with the indicated antibodies. Tail sequences of FGFR2[C1]; FGFR2[C1Δ34]; FGFR2[C2] and FGFR2[C3] are shown below. Densitometric bar graph represents 3 independent experiments. The error bars are presented as the standard deviation. **b** Proline-rich CT inhibits in vitro kinase activity. Recombinant KD-CT[C1], KD-CT[C2], KD-CT[C3] and control clone: KD-CT[C1Δ34] were incubated with ATP/Mg$^{2+}$ at 25 °C and quenched with 100 mM EDTA at different time points as indicated. The activation level was measured using an anti-pY657/658 antibody. Bottom: The densitometric analysis of kinase activity (KD-CT[C1] – blue; KD-CT[C2] – green; KD-CT[C3] – purple and KD-CT[C1Δ34] – red). **c** The affinities of CT[C1] to JM-KD[pY1] (red) and KD[pY1] (blue) determined using MST. CT[C1] was labeled with Atto488 dye then titrated with unlabeled JM-KD[pY1] and KD[pY1]. The error bars represent the standard deviation of 3 technical replications. **d** HSQC spectra of unbound $^1$H-$^{15}$N-labeled CT[C1] overlaid with KD[pY1]-bound CT[C1] at different ratio (black (0:1) to red (12:1)). Examples of peaks with high chemical-shift perturbations (CSPs) are shown by labels indicating the assignment of given peaks. CSPs chart of $^{15}$N-KD[pY1] titrated by CT[C1] was derived from $^1$H-$^{15}$N HSQC spectra. Large changes occur on both N-terminal and C-terminal residues of CT[C1]. **e** Wild type GST-CT[C1] and individual P to A mutants, the first 24 residues of CT (CT[C1Δ34]), the last 34 residues (CT[L34]), CT[C2] and CT[C1] D802F or D802W (to explore the importance of the charged acid group in binding) were used for a GST pulldown experiment with KD[pY1]. The symmetric dimerization of KD[pY1] at the concentration range used in this experiment (1 μM) was assumed to have negligible impact on binding of the various CT variants. Densitometric bar graph represents 2 independent experiments.

control (FGFR2[C1] with RFP) we see increasingly shorter lifetimes in the populations of C2 and C3 receptors respectively, indicating that, in the absence of growth factor, dimerization increases in response to reduction in the size of CT (Fig. 4c and Supplementary Fig. 4c). Interestingly the C3 isoform appears to be extensively membrane localized (zoomed inset panels Fig. 4c). So, the impact of CT appears to counter the dimerizing potential of JM. This respective negative and positive regulation of dimerization, and hence phosphorylation, by the regions N- and C-terminal of KD provides the opportunity for fine tuning of activation response.

**CT binds independently to JM.** We have demonstrated that CT binds to KD and affects dimerization and activity. But we also showed that CT binds more tightly to JM-KD[pY1] than to KD[pY1] alone (Fig. 3c). Thus, CT might be able to bind to both KD[pY1] and JM. This view is supported by the knowledge that only the N-terminal residues of JM are involved in the intermolecular latch interaction with KD (Fig. 1a), which would allow JM to preserve the asymmetric receptor dimer, whilst inhibiting interference in activity by binding to CT using its C-terminal residues. To investigate the possibility of a direct intramolecular interaction between JM and CT we first showed that binding of JM-KD[pY1] to both CT[C1] and CT[C2] was reduced as JM was truncated (Fig. 4d and Supplementary Fig. 4d respectively). Interaction with CT was much reduced on deletion of residues 429 to 449 which are outside the region previously shown to bind to KD, confirming that JM could maintain the latch interaction whilst simultaneously binding to CT. Using four mono-phosphorylated constructs; KD[pY1], JM-KD[pY1], KD[pY1]-CT[C1], and JM-KD[pY1]-CT[C1] in a pull-down assay with GST-CT, we showed that CT binds independently to JM-KD[pY1]. However, when including CT as part of the construct in both KD[pY1]-CT[C1] and JM-KD[pY1]-CT[C1] binding was abrogated (Fig. 4e). Thus, our data demonstrate that CT binds to JM-KD[pY1] through an intramolecular interaction, since including CT on the construct disrupts dimerization and blocks GST-CT binding.

We have shown that the presence of JM enhances the interaction with CT. Incubation of the same KD[pY1], JM-KD[pY1], KD[pY1]-CT[C1], and JM-KD[pY1]-CT[C1] with MBP-JM showed that, consistent with previous observations, JM was able to bind to KD[pY1] (Fig. 4f). JM also bound JM-KD[pY1], which, although dimerized through one JM latch, has a free KD for independent JM binding. Significant binding of JM to KD[pY1]-CT, but negligible binding of JM with the JM-KD[pY1]-CT[C1] construct was observed. These interactions of JM in the presence of CT could not occur if CT successfully competed with JM for binding to KD, but would require that JM can bind simultaneously with KD and CT. We measured direct binding

between JM and CT ($K_d = 20.2 \pm 2.92$ μM; Supplementary Table 6 and Fig. 4g). We also identified that the highest affinity sequence of JM that recognized CT includes residues $^{429}$VTVSAESSSSMNSN$^{442}$ (Supplementary Fig. 5a and Supplementary Table 7). Thus, the binding site on JM for CT is non-overlapping and C-terminal to the consensus sequence of JM that we showed is required for forming the intermolecular latch to KD[pY1], i.e., $^{414}$PAVHKLT$^{420}$, however it includes the VT site for FRS2 recruitment. Thus, as previously demonstrated with the cell-based assay (Fig. 3a), the presence of CT occludes FRS2 binding. This provides novel insight to a difference with FGFR1 which is constitutively bound to FRS2[37].

To map the interaction between JM onto CT[C1] we titrated unlabeled JM into $^{15}$N-labeled CT[C1]. Using NMR the binding site can be seen to incorporate residues between V798 and S819 of CT (Fig. 4h). Using a series of short peptides derived from CT we demonstrated that the proline-rich sequence from CT binds to JM (Supplementary Table 7 and Supplementary Fig. 5b), and $^{808}$PCLPQYPH$^{815}$ sequence is necessary for CT to bind to JM. Importantly, this is the same sequence that binds to both KD and to the GRB2 CSH3 domain[30]. Thus, CT mediates three modes of receptor regulation. The potential previously unrecognized importance of proline-rich motifs on RTK CTs is again emphasized here.

**JM and CT regions combine to regulate kinase activity.** Finally, we examined how the presence of JM and CT regulate FGFR2 signaling using KD[pY1]; JM-KD[pY1]; KD[pY1]-CT[C1]; and JM-KD[pY1]-CT[C1]. The kinase dead JM-KD[K518I]-CT was used as a substrate for the different constructs (Fig. 5a). The activity is slightly enhanced by the presence of JM in JM-KD[pY1] compared to KD[pY1]. This is consistent with JM stabilizing the active asymmetric dimer. Conversely, the presence of CT in KD[pY1]-CT[C1] dramatically inhibits kinase activity through the previously observed direct interaction with KD and resulting inhibition of dimerization. The presence of JM and CT in JM-KD[pY1]-CT[C1] shows medium activity. This underscores the regulatory role of the interplay between the two peripheral regions of the receptor in sustaining the active intermediate state through modulation of kinase activity, both by being able to stabilize the asymmetric dimer conformation and inhibit the enzyme through blocking dimerization. These data mirror the experiments on the activation of the K*sam* truncated FGFR2 isoforms and the exon truncations in the human gene, i.e., in the absence of CT the down-regulation of activity of the intact receptor is disrupted (Fig. 3a).

As the receptor becomes progressively phosphorylated the control imposed by CT needs to be downregulated to enable

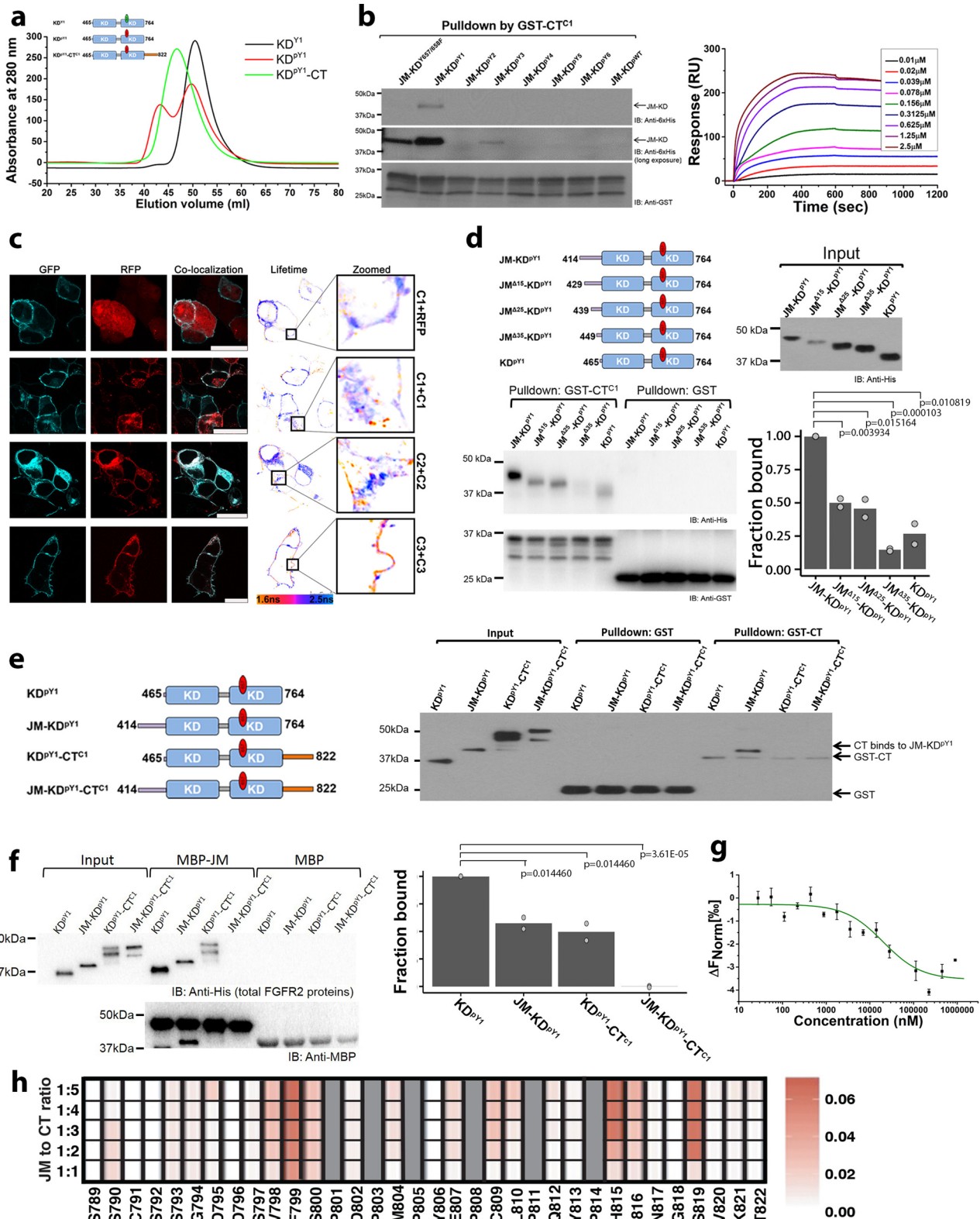

access of downstream signaling proteins. We used a pull-down experiment to reveal the mechanism for this release. GST-CT which was phosphorylated on its available tyrosine residues, pCT$^{C1}$, was unable to pulldown JM-KD as it became progressively phosphorylated, i.e., JM-KD$^{pY1}$-JM-KD$^{pY6}$ (Fig. 5b) Thus, as the receptor pY load increases, the phosphorylated CT is less able to bind intramolecularly, making it available for recruitment of downstream effector proteins.

We showed that residues in JM sequence 429 to 449 play a role in interacting with CT (Fig. 4d), and hence the presence of CT would occlude the $^{429}VT^{430}$ binding motif[37] for the FRS2 phosphotyrosine binding domain (PTB). This inhibition of access of FRS2 to JM by CT was demonstrated where significantly less FRS2 PTB domain was pulled down in the presence of CT, JM-KD$^{pY1}$-CT$^{C1}$ compared to JM-KD$^{pY1}$-CT$^{C3}$ (Fig. 5c). JM-KD$^{pY1}$-CT$^{C2}$ showed an intermediate level of interaction consistent with

**Fig. 4 CT binding to KD$^{pY1}$ disrupts the formation of asymmetric dimer. a** Dimerization status of FGFR2 kinases in the presence or absence of CT. KD$^{pY1}$ (red curve) includes a dominant population of dimers at 60 μM in SEC (HiPrep Sephacryl 26/60 S-100 HR), whereas the unphosphorylated KD (black curve) prevails as a monomer. The mono-phosphorylated KD$^{pY1}$-CT construct (green curve) also exists as a monomer in solution. **b** Phosphorylation of JM-KD affects CT binding. GST-CT$^{C1}$ was used to pulldown the progressively increasing phosphorylation of JM-KD constructs. The pulldown indicates that JM-KD$^{pY1}$ was the highest affinity binding partner for GST-CT$^{C1}$. This interaction is verified using SPR. See Supplementary Table 5 for the affinity measurements. **c** Dimerization of FGFR2 by FLIM analysis of the FRET between the FGFR2-GFP and FGFR2-RFP. First panel: Reference lifetime measurements between FGFR2-GFP and RFP-alone, control for no interaction. The mean lifetime is centered around ~2.1 ns (dark blue), which corresponds to the mean lifetime for isolated CFP alone. Second panel: Dimerization of C1 showing a measurable left shift with of the molecule showing FRET above the control. Note that most interactions are seen in the intracellular vesicles. Third panel: Dimerization of C2. 16% of molecules on plasma membrane showing dimerization above the control threshold (orange). Fourth panel: Dimerization of C3, as with C1, 26% of the molecules are showing interaction (orange) however unlike C1, almost all of the interactions are on the plasma membrane. Inserts with arrows showing exquisite separation of dimeric and non-dimeric FGFR2-C3 on the plasma membrane. Representative of 3 independent experiments. 10–15 cell images were taken for each isoform in each independent experiment. Scale bar = 25 μm. **d** GST-CT$^{C1}$ was used to pull down five mono-phosphorylated constructs of JM-KD$^{pY1}$ with progressively truncated JM (JM-KD$^{pY1}$, JM$^{Δ15}$-KD$^{pY1}$, JM$^{Δ25}$-KD$^{pY1}$, JM$^{Δ35}$-KD$^{pY1}$, and KD$^{pY1}$). The presence of the intact JM enhances the interaction with GST-CT$^{C1}$. Densitometric bar graph represents 2 independent experiments. **e** A GST-CT$^{C1}$ pulldown of different FGFR2IIIb mono-phosphorylated proteins that include the presence or absence of JM and/or CT (KD$^{pY1}$; JM-KD$^{pY1}$; KD$^{pY1}$-CT, and JM-KD$^{pY1}$-CT), shows that the presence of JM, but not CT, promotes the interaction between kinase domain and GST-CT$^{C1}$. The presence of CT inhibits the GST-CT$^{C1}$ interaction, indicating CT binds through an intramolecular interaction. **f** A MBP-JM pulldown of different FGFR2IIIb mono-phosphorylated proteins (as described in Fig. 4e), shows that the presence of JM does not block JM binding suggesting that JM of one protomer binds to the other in the mono-phosphorylated dimers (previously identified for KD and JM-KD). The latch to the protomer in the asymmetric dimer leaves an available JM binding site. The presence of CT (in KD-CT$^{C1}$ and JM-KD$^{pY1}$-CT$^{C1}$) reduces JM binding. Densitometric bar graph represents 2 independent experiments. **g** MST measurement of JM binding to CT. A two-fold serial dilution of CT was titrated into JM which was labeled with Atto 488. The error bars represent standard error of the mean from 3 technical replications. **h** NMR titration of JM titrated into $^{15}$N-labeled CT using a red-to-white gradient, where white represents the weakest CSP and red depicts the strongest CSP. Proline residues are not visible in this experiment (shown in gray).

the proline-rich motif present in this isoform binding with lower affinity to the FRS2 cognate site. We also measured different FGFR2 isomers binding to FRS2 using BLI. In the absence of the intact CT (C3 isoform) a significantly increased amount of FGFR2 protein bound to the PTB domain compared with the C1 and C2 isoforms (Fig. 5d).

Using an in vitro kinase assay we were able to demonstrate that the phosphorylation of FRS2 by FGFR2 is affected by CT in the different isoforms. Immunoblotting showed that the C3 isoform has the highest kinase activity toward FRS2, whereas the C1 isoform has the lowest (Fig. 5e). This further suggested that CT$^{C1}$ isoform can interact with JM which contains the $^{429}$VT$^{430}$ motif and inhibit the recruitment and phosphorylation of FRS2. This observation explains why the C3 isoform has higher FRS2-mediated downstream signaling activity and exhibits uncontrolled activation leading to oncogenic outcome in the active intermediate state.

## Discussion

As has been seen with other RTKs, FGFR2 cycles through a series of states on going from the isolated, dephosphorylated monomeric state to the fully phosphorylated, signaling-competent state. Our data provide exquisite insight to these defined states. As with all studies that report on the complex series of events that lead to RTK phosphorylation, we investigated a series of snapshots of structural states which show possible interactions and juxtapositioning of the various components prior to growth factor binding; under this condition FGFR2 prevails in a stable, monophosphorylated state. In addition to the unphosphorylated structure, there are numerous FGFR structures that provide detail on 'trans-phosphorylation' events on different tyrosines[23,41,42]. Linking of these snapshots into an animated series permits a full understanding of the progression of events from the unphosphorylated state, through the intermediate monophosphorylated state (unliganded, basal RTKs), to the active trans-phosphorylating state (ligand-bound state) and how each one provokes the next. Our data have highlighted the monophosphorylated active intermediate JM-KD$^{pY1}$-CT state as the most important frame in the animation of the progress from

inactive to signaling receptor. This state represents a major checkpoint because the KD is active, but signal transduction is inhibited. Our observations show how, in this state, the receptor is highly regulated by the interactions of both JM and CT whilst being 'primed' for full activation on growth factor binding.

Data on the binding of JM to KD (Fig. 1c, d) and the effect of JM on the dimerization (Fig. 2b) reveal that JM appears to inhibit the direct interaction between KDs and promote dynamic inter-locution between the active domains in an asymmetric dimer. It also juxtaposes KDs in such a way to prevent progressive oligo-merization of the domains (i.e., 'daisy chain' formation[2,43]) allowing only dimers to form. Tyrosine phosphorylation of CT provides sites for downstream effector protein recruitment. Thus, CT needs to be able to access the catalytic site of KD. However, to avoid unregulated phosphorylation, and hence aberrant signal transduction, the positioning of CT needs to be strictly controlled. This is achieved by independent binding to KD, JM or GRB2[30] depending on the requirements of the receptor at a given point in the cycle. Truncation of this region up-regulates the kinase (Fig. 3c, b) and hence provides a definitive rationale for the elevated proliferative signaling in the oncogenic K*sam* deletions and exon truncation.

The ability of CT to bind mutually exclusively to both KD and JM suggests that it can adopt two distinct conformations which have opposing impact on kinase activity which would operate independently and at different points in the receptor up-regulation process. 1) Binding of CT to KD$^{pY1}$ results in an intramolecular auto-inhibitory conformation which sustains the monomeric state of the mono-phosphorylated receptor. This is expected to occur in the absence of growth factor stimulation to prevent further up-regulation. 2) The CT of the enzyme-like KD binds intramolecularly to the JM, which is also acting as the latch to the substrate-like KD of the asymmetric dimer. Thus, the CT is effectively isolated from blocking the kinase activity and receptor dissociation. In this conformation the CT of the substrate-like protomer is free to become phosphorylated. We propose that this occurs when the receptor is bound to extracellular growth factor and full activation is stimulated. Thus, extracellular ligand-mediated stabilization of receptor dimers promotes JM latch

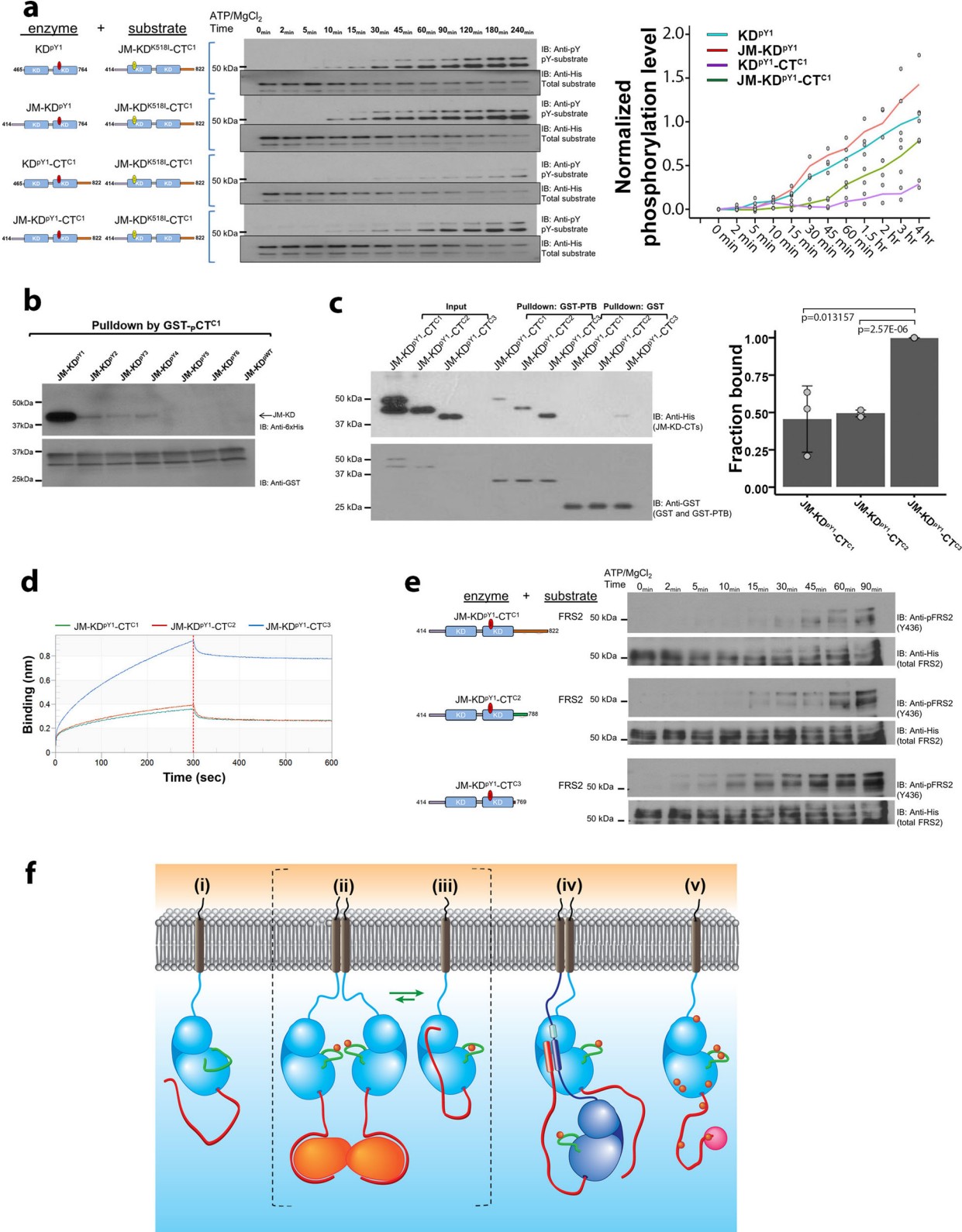

formation and allows CT to relocate from the kinase inhibitory/ dimer disrupting position on KD$^{pY1}$, to binding to JM. The direct interaction between two peripheral regions adds a level of control to the kinase output not previously observed for RTKs.

The herein study provides a mechanistic model for the progression of the cytoplasmic region of FGFR2 moving from the dephosphorylated, inactive monomeric state, via the mono-

phosphorylated active intermediate, through to the fully active state (Fig. 5f). Unphosphorylated receptor diffuses through the plasma membrane (Fig.5f i). Random self-association of monomeric FGFR2 can lead to as much as 20% dimer under basal conditions[44]. GRB2 can stabilize FGFR2 dimers, resulting in trans-phosphorylation of the A-loop Y657 (Fig. 5f ii). We hypothesize that the presence of GRB2 also restricts dynamic

**Fig. 5 JM and CT combine to regulate KD. a** Kinase activity is controlled by both JM and CT. $KD^{pY1}$, $JM-KD^{pY1}$, $KD^{pY1}-CT^{C1}$, and $JM-KD^{pY1}-CT^{C1}$, were incubated with kinase-dead $JM-KD^{K518I}-CT^{C1}$ in a 1:1000 ratio in the presence of $ATP/Mg^{2+}$ and quenched with EDTA at different time points as indicated. The phosphorylation of $JM-KD^{K518I}-CT^{C1}$ was measured using a pY99 antibody. Densitometric line graph represents 2 independent experiments. The mean of the two replicates (shown as points) is represented by the colored lines. **b** Phosphorylation of CT ($pCT^{C1}$) reduces its binding to KD with higher phosphorylation order. $GST-CT^{C1}$ was phosphorylated by $JM-KD^{WT}-CT^{C1}$ and used for a GST pulldown assay with different phosphorylated JM-KD (JM-$KD^{pY1}$ – $JM-KD^{pY6}$ and wild type JM-KD). **c** GST-FRS2 PTB domain was used precipitate the following His-tagged constructs representing the mono-phosphorylated isoforms of FGFR2IIIb; $JM-KD^{pY1}-CT^{C1}$, $JM-KD^{pY1}-CT^{C2}$ and $JM-KD^{pY1}-CT^{C3}$. The presence of the intact CT in the C1 isoform inhibits the interaction of FRS2 with its cognate site on JM. Densitometric bar graph represents 3 independent experiments. The error bars are presented as the standard deviation. **d** BLI measurement of GST-FRS2 PTB binding to $JM-KD^{pY1}-CT^{C1}$, $JM-KD^{pY1}-CT^{C2}$ and $JM-KD^{pY1}-CT^{C3}$. The GST-PTB domain from FRS2 was immobilized on the sensor and was exposed to $2.6\,\mu M$ of the cytoplasmic region of each of the FGFR2 isoforms. After 300 s the chip was washed. The sensorgrams clearly show that over the time course up to 300 s (prior to the washing step; dotted line), in the absence of CT (C3 isoform) a significantly increased amount of FGFR2 protein binds to the PTB domain compared with the C1 and C2 isoforms. **e** The FGFR2 isoforms ($JM-KD^{pY1}-CT^{C1}$, $JM-KD^{pY1}-CT^{C2}$ and $JM-KD^{pY1}-CT^{C3}$) were incubated with FRS2 protein in a 1:100 ratio in the presence of $ATP/Mg^{2+}$ and quenched at different time points as indicated. The phosphorylation of FRS2 was measured using an anti-pFRS2 (Y436) antibody. **f.** i: In the absence of stimulation the unphosphorylated FGFR2 (light blue, JM light blue line, CT red line) can exist as a monomer freely diffusing through the plasma membrane. ii: Random collision of FGFR2 results in dimer formation. Dimeric GRB2 (orange) is recruited via a proline-rich sequence on CT into a heterotetramer[30]. This stabilizes the mono-phosphorylated active A-loop (green line) tyrosine residues (red circles) on KD, but signaling is stalled by the presence of GRB2 on CT. iii: the mono-phosphorylated KD also provides a strong binding site for CT. $CT-KD^{pY1}$ interaction results in the release of GRB2. The presence of CT on the KD prevents JM-mediated formation of asymmetric dimer and hence aberrant up-regulation. Active intermediate states ii: and iii: are in equilibrium, the concentration of the states is dependent on GRB2 concentration and the ability of GRB2 to compete with the intramolecular interaction with KD for binding to CT. iv: Binding of extracellular growth factor co-localizes two receptors into the active, asymmetric dimeric conformation. This is sustained by the interaction of JM from the enzyme-like receptor (dark blue) with KD of the substrate-like receptor (light blue). The sequence on JM which binds to KD (light blue thick line) is immediately proximal to a sequence (dark blue thick line) which binds in an independent interaction to CT of the enzyme-like receptor. This binding site (red thick line) includes the proline-rich motif that recognizes a site on KD and GRB2. Thus, JM-CT interaction blocks auto-inhibition and GRB2 recruitment. This ensures that the active state is prolonged. v: Prolonged activity of the dynamic asymmetric dimer results in increasing phosphorylation of KD and CT. As the pY burden increases the dimerization affinity between KDs reduces until they fully dissociate. The phosphorylated KD abrogates the inhibitory intramolecular binding of CT and the recruitment of GRB2. The receptor is therefore available for recruitment of downstream effector proteins (magenta). In the presence of growth factor the receptor would be expected to prevail as a dimer, however on dissociation of the growth factor the fully phosphorylated form may be expected to be a monomer.

motion of $KD^{pY1}$, potentially stabilizing the symmetric dimer conformation (Supplementary Fig. 2d). This hypothesis is consistent with symmetric dimer seen in the inactive states of other RTKs[45]. Our observations also demonstrate that CT binds intramolecularly to $KD^{pY1}$ and more tightly to $JM-KD^{pY1}$ (Fig. 3c). This interaction is mutually exclusive of binding of GRB2 (Fig. 5f iii). The presence of CT bound to $KD^{pY1}$ inhibits activity by blocking asymmetric dimerization. Thus, the mono-phosphorylated state, which is the checkpoint prior to full receptor activation, is tightly regulated either by the binding of GRB2 or the interaction with CT.

Exposure of cells to FGF and concomitant receptor conformational change, results in JM of one protomer in the dimer latching onto KD of the other. In this way the former becomes the designated enzyme-like receptor, whilst the latter becomes the substrate-like receptor, both being held in a moderate affinity, dimeric conformation (Fig. 5f iv). Since the KD is already in its mono-phosphorylated state it is primed for JM binding and adoption of the asymmetric dimer conformation. The unrestricted activity of the enzyme-like receptor requires that CT is not able to bind to, and hence down-regulate KD of this protomer. This state is achieved through the binding of CT to the available intramolecular JM recognition sequence. The progressive weakening of interactions of the peripheral regions as a result of progressive phosphorylation increases the dynamic interplay between protomers permitting easier access to phosphorylation sites and alternation of the enzyme-like and substrate-like states between the molecules.

Progressive phosphorylation of KD also weakens the interaction with JM (Fig. 1c) and results in dimer dissociation (Fig. 2a) leaving the receptor in a highly phosphorylated state whereby it can recruit downstream effector proteins (Fig. 5f v). Dissociation of the phosphorylated FGFR2 leaves it exposed to phosphatase activity which ultimately returns it to its initial unphosphorylated

state (Fig. 5f i). Clearly the controlled activation cycle of FGFR2 would be affected by the impact of additional factors such as endocytosis[39], fluctuations in GRB2 concentration and phosphatase concentration[27]. However, the importance of both peripheral regions in influencing the self-association, and the dimeric conformation underscores how the receptor is tightly regulated to avoid aberrant signaling.

The evolution of different RTKs appears to endow crucial features enabling idiosyncratic regulation and commitment to defined downstream outcomes. Comparison with EGFR throughout this report has emphasized this point. A major feature of FGFR2 regulation focuses on the role of proline-rich sequences in the peripheral regions: sequences which appear frequently in RTKs but have not been investigated for regulatory impact. The *K-sam*II truncations show that when these regulatory features are perturbed pathogenicity can result in uncontrolled cellular signaling (Supplementary Fig. 5c). Thus, understanding of the roles of the peripheral region interactions will suggest alternative routes for therapeutic intervention outside the currently well-trodden path of inhibition of kinase activity.

## Materials and methods

**Cell culture**. HEK293T cells were maintained in DMEM (Dulbecco's modified Eagle's high glucose medium) supplemented with 10% (v/v) FBS (fetal bovine serum) and 1% antibiotic/antimycotic (Lonza) in a humidified incubator with 10% $CO_2$.

**Protein expression and purification**. All MBP-tagged, GST-tagged and 6xHistidine-tagged fusion proteins were expressed and purified from BL21(DE3) cells. A single colony was used to inoculate 100 mL of LB which was grown overnight at 37 °C. 1 L of LB was inoculated with 10 mL of the overnight culture and allowed to grow at 37 °C until the $OD_{600}$ reaches 0.8 at which point the culture was cooled down to 20 °C. Expression was then induced with 0.5 mM IPTG and the culture was grown for a further 12 h before harvesting by centrifugation. Cells were re-suspended in 20 mM Tris, 150 mM NaCl, 10% glycerol, pH 8.0 in the presence of protease inhibitors and lysed by sonication. Insoluble material was removed by centrifugation (40,000 g at 4 °C for 60 min). The soluble fraction was

applied to an appropriate affinity column (Amylose column for MBP-tagged proteins, GST column for GST-tagged proteins and Talon column for His-tagged proteins). Following a wash with 10 times column volume of wash buffer (20 mM Tris, 150 mM NaCl, pH 8.0), the protein was eluted from the column with elution buffer (the washing buffer supplemented with 20 mM maltose for the MBP-tagged proteins; a supplement of 20 mM reduced glutathione for the GST-tagged proteins; a supplement of 150 mM imidazole for the 6xHis-tagged proteins) and was concentrated to 5 mL and applied to a Superdex 75 gel filtration column equilibrated in a buffer containing 20 mM HEPES, 150 mM NaCl and 1 mM TCEP pH 7.5. Analysis for protein purities by SDS-PAGE showed greater than 98% purity. For CT (CT$^{C1}$ and CT$^{C1\Delta34}$, GST-tagged) production and JM-KD$^{PY1}$-CT$^{C1}$ (for crystallography, 6xHis-tagged), 1 unit of thrombin (Sigma T6884) was used to cleave 1 mg of recombinant proteins at 4 °C for overnight. After cleavage, Benzamidine Sepharose 4 Fast Flow beads (GE) were used to remove thrombin. GST-Tag/His-Tag and uncut proteins were removed by passing protein solution through a GST or Talon column.

Expression of $^{15}$N-labeled proteins for NMR titrations was done as previously described[46]. For expression in 100% D$_2$O, this procedure was modified by pre-growing the culture in a small volume of 100% D$_2$O prior to expression over 20 h.

### Nuclear magnetic resonance (NMR) spectroscopy

*General information.* Titration experiments were carried on Bruker Avance III 750 MHz NMR spectrometer, equipped with $^1$H-optimized triple resonance NMR 5 mm TCI-cryoprobe.

NMR data was processed using NMRPipe[47] and further analyzed with CcpNmr Analysis software package[48] available locally and on NMRBox platform[49]. Chemical shift perturbations (CSPs) were calculated from the chemical shifts of backbone amide $^1$H ($\Delta\omega_H$) and $^{15}$N ($\Delta\omega_N$) using the following equation: CSP = $\sqrt{\Delta\omega_H^2 + \left(0.154\Delta\omega_N^2\right)}$[50].

*CT backbone assignment.* To obtain backbone assignment of CT, polypeptide was expressed in isotopically labeled media as described above. Spectra of CT were recorded using 300 μM sample in the same HEPES buffer. Standard Bruker library together with BEST versions[51] of amide transverse relaxation optimized spectroscopy (TROSY)[52] of 3D backbone resonance assignment pulse sequences (HNCA, HNCOCA, HNCACB, CACBCONH, HNCO and HNCACO) were applied to collect high resolution spectra. In order to shorten acquisition time, Non-Uniform Sampling (20–30%) was routinely used.

*CT titration with KD.* The $^{15}$N-labeled CT sample concentrated to 300 μM in HEPES buffer was titrated with unlabeled KD. Amide spectra were recorded at 25 °C using hsqcetfpf3gpsi pulse sequence from Brüker library at 1:0.5, 1:1, 1:2, 1:3, 1:4, 1:8 and 1:12 molar ratios.

### X-ray crystallography

Crystals of JM-KD$^{PY1}$-CT$^{C1}$ were obtained using the hanging-drop vapor diffusion method, mixing equal volumes of protein with reservoir solution and equilibrating over this reservoir at 20 °C for 2 weeks. The reservoir solution contained 100 mM Tris, 160 mM TMAO, 20% PEG2000 at pH 8.6.

For cryoprotection, crystals were transferred in the crystallization buffer supplemented by 20% Glycerol. X-ray diffraction data sets were collected from frozen single crystals at the Advanced Light Source (Berkley, CA, USA, beamline 8.3.1) and processed with the program Elves. A molecular replacement solution was obtained using the BALBES molecular replacement pipeline and the crystal structure PDB code 2PSQ. Iterative model rebuilding and refinement was performed by using the program COOT, REFMAC5 and PDB_REDO against the data set. Structural figures were made using PyMol.

See Supplementary Table 3 data collection and refinement statistics.

### Mutation of FGFR2 proteins

Standard site-directed mutagenesis was carried out to mutate tyrosine residues into phenylalanine on KDs to mimic the sequential phosphorylation pattern of KD (KD$^{PY1}$ to KD$^{PY6}$; see schematic Fig. 1C and Supplementary Fig. 2A. For the FGFR2 IIIb isoform this sequence is; pY657, pY587, pY467, pY589, pY658 and pY734, adapted from the IIIc isoform[23]). The same methods were also used to mutate proline residues on the C-terminal tail in this study.

### In vitro dephosphorylation *and* phosphorylation of purified proteins

Calf Intestinal alkaline phosphatase (CIP, New England Biolabs) was conjugated on UltraLink Biosupport beads (Thermo Fisher Scientific). CIP-beads were mixed with purified protein solution and rotated gently at 4 °C for overnight to remove phosphate group in solution. After dephosphorylation, protein solution and CIP-beads were separated by centrifugation. Dephosphorylation level was examined by western blotting.

Purified FGFR2 proteins were phosphorylated by incubating with 5 mM ATP and 10 mM MgCl$_2$. The phosphorylation reactions were quenched by adding EDTA (prepared in 10 mM HEPES, pH 7.5) to a final concentration of 100 mM.

Proteins were analysed by SDS-PAGE and western blot to study the phosphorylation status.

### Transient cell transfection with plasmids

30 min before transfection, cells were harvested and resuspended in antibiotic-free medium. Transfection was carried out using Metafectene (Biontex Cat#: T020) according to manufacturer manual.

### Cell signaling studies

For mammalian cell studies, cells were starved for 16 h, and left unstimulated or stimulated with 10 ng/ml FGF7 ligand (R&D Systems Cat#: 251-KG/CF) at 37 °C. After stimulation, medium was removed and cells were put on ice and immediately lysed by scraping in ice-cold lysis buffer supplemented with protease inhibitor (Calbiochem) and phosphatase inhibitor (1 mM sodium orthovanadate (NaVO$_3$), and 10 mM sodium fluoride (NaF). Cells were cleared by centrifugation and the supernatants were subjected to immunoblotting using the BioRad protein electrophoresis system. The intact gel was transfer to PVDF membrane for probing with different antibodies.

Phospho-protein blots were stripped with stripping buffer (Millipore) and re-probed with total protein antibodies. Antibodies were from: Anti-Phospho-FGF Receptor (Tyr653/654) Rabbit polyclonal, Cell Signaling Technology Cat#:3471; Anti-Phospho-FRS2-α (Tyr436) Rabbit polyclonal, Cell Signaling Technology Cat#: 3861; Anti-Phospho-p44/42 MAPK (Erk1/2) (Thr202/Tyr204) Rabbit monoclonal, Cell Signaling Technology Cat#: 4370; Anti-p44/42 MAPK (Erk1/2) Rabbit monoclonal, Cell Signaling Technology Cat#: 4695; Anti-α-Tubulin Rabbit polyclonal, Cell Signaling Technology Cat#: 2144; Anti-GST Rabbit polyclonal, Cell Signaling Technology Cat#: 2622; Anti-FGFR2 Mouse monoclonal, Santa Cruz Biotechnology Cat#: sc-6930; Anti-Phospho-Tyr Mouse monoclonal, Santa Cruz Biotechnology Cat#: sc-7020; Anti-6xHis Mouse monoclonal, Takara Cat#: 631212.

### Pulldown and western blots

For immunoblotting, proteins were separated by SDS-PAGE, transferred to PVDF membranes and incubated with the specific antibodies. Immune complexes were detected with horseradish peroxidase conjugated secondary antibodies and visualized by enhanced chemiluminescence reagent according to the manufacturer's instructions (Pierce).

For pulldown experiments, 100 ug of protein was prepared in 1 ml volume. MBP-tagged or GST-tagged proteins immobilized on Amylose beads (GE Healthcare Life Science) or Glutathione Sepharose (GE Healthcare Life Science) was added and incubated at 4 °C overnight with gentle rotation. The beads were then spun down at 4000 rpm for 3 min, supernatant was removed and the beads were washed with 1 ml lysis buffer. This washing procedure was repeated five times in order to remove non-specific binding. After the last wash, 50 μl of 2x Laemmli sample buffer were added, the sample was boiled and subjected to SDS-PAGE and western blot assays.

### Fluorescence resonance energy transfer (FRET)

Recombinant GFP-JM-KD$^{PY1}$-CT$^{C1}$ (donor) and RFP-JM-KD$^{PY1}$-CT$^{C1}$ (acceptor) proteins (1 μM) were used for In vitro steady-state FRET analysis. The changes of donor emission (510 nm) upon dimer formation or dimer disruption upon the addition of CT were recorded at 25 °C.

### Quantitative imaging FRET microscopy

HEK293T cells 24 h after transfection were seeded onto coverslips and allowed to grow for a further 48 h then fixed by addition of 4% (w/vol) paraformaldehyde, pH 8.0, 20 min at room temperature. Cells were then washed six or seven times with PBS, pH 8.0 and mounted onto a slide with mounting medium (0.1% p-phenylenediamine/ 75% glycerol in PBS at pH 7.5–8.0) and curated for 3–4 h before imaging. FLIM images were captured using a Leica SP5 II confocal microscope. Atto488 was excited at 900 nm with titanium–sapphire pumped laser (Mai Tai BB, Spectral Physics) with 710–920 nm tunability and 70 femtosecond pulse width. Becker & Hickl (B&H) SPC830 data and image acquisition card was used for time-correlated single photon counting (TCSPC). Electrical time resolution 8 Pico seconds with a pixel resolution of 512 × 512. Data processing and analysis were done using B&H SPC FLIM analysis software. The fluorescence decays were fitted with a single exponential decay model.

### Microscale thermophoresis (MST)

Binding affinities were measured using the Monolith NT.115 (NanoTemper Technologies, GmbH). Proteins were fluorescently labeled with Atto488 according to the manufacturer's protocol. Labeling efficiency was determined to be 1:1 (protein:dye) by measuring the absorbance at 280 and 488 nm. A 16 step dilution series of the unlabeled binding partner was prepared and mixed with the labeled protein at 1:1 ratio and loaded into capillaries. Measurements were performed at 25 °C in a buffer containing 20 mM HEPES, 150 mM NaCl, 1 mM TCEP and 0.01% Tween 20 at pH7.5. Data analysis was performed using Nanotemper Analysis software, v.1.2.101 and was plotted using Origin 7.0. All measurements were conducted as technical duplicates or triplicates. For the experiments employed to measure dimerization KD values are referred to as 'apparent' because, based on the differential concentrations, the fitting model assumes labeled are bound to unlabeled polypeptides.

**Surface plasmon resonance (SPR)**. SPR experiments were carried out using a BIAcore T100 instrument (GE Healthcare). CT[C1] were immobilized on CM4 chips according to the standard amine coupling protocol. Briefly, carboxymethyl groups on the chip surface were activated with a 1:1 mixture of N-ethyl-N-(dimethya-minopropyl) carbodiimide (EDC) and N-hydroxysuccinimide (NHS). Proteins were diluted in 20 mM HEPES, pH 6.5 and injected over the activated chip surface. The unbound chip surface was blocked using ethanolamine. Proteins were immobilized to approximately 200 response units. Different concentrations of analytes were injected over the immobilized chips at a flow rate of 30 μl/min. The sensor surface was regenerated by injection of 30 μl of 0.1% SDS and 60 μl of 500 mM NaCl. Reference responses were subtracted from flow cells for each analyte injection using BiaEvaluation software. The resulting sensorgrams were anaylsed to determine the kinetic parameters. Raw data shows a rise in signal associated with binding followed by a diminished signal after application of wash buffer.

**Bio-layer interferometry (BLI)**. BLI experiments were performed using a FortéBio Octet Red 384 using Anti-GST sensors. Assays were done in 384 well plates at 25 °C. Association was measured by dipping sensors into solutions of analyte protein (FGFR2 proteins) for 125 s and was followed by moving sensors to wash buffer for 100 s to monitor the dissociation process. Raw data shows a rise in signal associated with binding followed by a diminished signal after application of wash buffer.

**Peptides**. [407]Juxtamembrane region[462], [407]KPDFSSQPAVHKLT[420], [414]PAVHKLTKRIPLRRQVT[430], [429]VTVSAESSSSMNSN[442], [439]MNSNTPLVRITTRL[452], [449]TTRLSSTADTPMLA[462], [801]PDPMPYEP[808], [801]PDPMPYEPCLPQYPH[815], [808]PCLPQYPHINGSVKT[822], [801]PDPMPYEPCLPQYPH[815], [808]PCLPQYPH[815], [808]PCLPQYPHINGS[819], [815]HINGSVKT[822], [804]MPYEPCLP[811]. All peptides were purchase from Genscript.

**Statistics and reproducibility**. All data were expressed as mean and standard deviation or standard error of the mean as indicated in figure legends. *P* values were determined by the Student's t test.

**Reporting summary**. Further information on research design is available in the Nature Portfolio Reporting Summary linked to this article.

## Data availability

The Accession Code for the crystal structure on the PDB is PDB6V6Q. The PDB validation report is available at https://www.rcsb.org/structure/6V6Q The authors declare that all data supporting the findings of this study are available within the article and its supplementary information files. All data needed to evaluate the conclusions in the paper are present in the paper or the Supplementary Data 1–5.

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

## Acknowledgements

This work was funded in part by CRUK grant C57233/A22356 awarded to J.E.L. The research by S.T.A. supported by funding from King Abdullah University of Science and Technology (KAUST). Z.A. is supported by National Institutes of Health (NIH) grant R01 CA200231 and Cancer Prevention Research Institute of Texas (CPRIT) grant RP180813. We acknowledge SOLEIL for provision of synchrotron radiation facilities (proposals nr. 20181104 and 20190107) and we would like to thank J. Perez and A. Thureau for assistance in using the beamline SWING. We thank A. Stainthorp for helpful discussion and comments. We thank Dr A. Kalverda (The Astbury Structural Biology Laboratory, University of Leeds) for the help on NMR data collection and analysis.

## Author contributions

All authors devised and performed experiments based on biochemistry, biophysics, cell/molecular biology (C.–C.L.), NMR spectroscopy (L.W.), X-ray crystallography and interpretation of structural data (G.P.-M., S.T.A), biochemistry, biophysics (K.M.S.), microscopic imaging (Z.A.). The manuscript was written by J.E.L and C.-C.L.

## Competing interests

The authors have no competing interests to declare.
