## [Peer Review File · Communications Biology]

Reviewers' comments:

Reviewer #1 (Remarks to the Author):

Lin et al in their manuscript "The combined action of the intracellular regions regulate oncogenic FGFR2 kinase activity" present an argument for the independent and combined regulatory actions of the JM (juxtamembrane) and CT (C-terminal) peptides upon the central enzymatically active KD (kinase domain) of the RTK (receptor tyrosine kinase) FGFR2 (fibroblast growth factor receptor II). Whilst their report does include some evidence in support of (rather consistent with) their central thesis, much presented data does not unequivocally support their conclusions, or the presented data is only tangentially related to their argument. This manuscript needs a substantial re write and re organization to make it more readable and to focus more on their main points.

Main issues:

1. Data for sequential and progressive phosphorylation affecting JM binding to KD

The data presented from a pull down in 1c simply shows that pY1 is preferred to all other mutants. Why is there no WT here? Why not show statistical significance? Why Ponceau stain labeled MBP-JM? Shouldn't this be MBP-JM higher band, and MBP alone lower band? Fig 1d seems to be the experiment that attempts to show quantitative differences between mutants. However, the fits are very poor. And the relative dFnorm for the different mutants is all different. Why doesn't each have same max change? They each have same mass and fold. More importantly, the fits are very poor. Apart from pY1 most can be fit to a straight line and do not approach a plateau at high concentration. Together this makes me question the stability of all the other mutants. Why not do SPR for the binding assays. CT binding in Supp Fig 6b much more clear.

I would remove fig 1c and 1d from the main figs and place in supplement, or remove altogether. The data that is in supplement 1d is more convincing for binding. However, the fit is poor 414-430. In fact I am unsure how that was fit. It completely misses the high concentration point, and there is no indication of a plateau. My conclusion by looking at the data is that 407-420 binds with the highest affinity. And 414-430 is weaker by an order of magnitude. However, additional controls need to be employed to show specificity. One worry with studies of small peptides is to ensure that it is not a non specific hydrophobic interaction. Here we see a direct patten to binding with hydrophobicity. 407-420 as the only peptide with a bulky Phe. And 414-430 with 5 small hydrophobics. And the non binding peptides with substantially less hydrophobic character. What is necessary is a scramble peptide experiment of the binders to show specificity.

For figure 1e-f the authors use NMR to show interacting residues on the kinase domain. However the figures add no value for the reader. The whole KD is not shown and key residues identified by NMR are missing (N terminal prior to 468). Since nothing mechanistic can be gleaned from this, it can be moved to supplement. And just stated as N-term residues up to 475 showed most shift. Were these NMR interactions tested with binding? Repeat experiment of supplement 1d with KD that lacks this 'critical' N-term. Or specific mutations of NMR identified residues.

2. SAXS data is not robust for showing the authors suggestions of different species

The dimerization constant presented for pY1 dimerization of 100nM (Fig 2c) means that at the concentrations used for SAXS there should be near complete dimerization. Showing residuals of plots to different imposed models does not confirm the symmetric or asymmetric orientations. The authors should generate bead models and dock, or rather use negative stain EM. If the authors really wanted to show asymmetric vs symmetric orientations they should use the classic experiments of donor receiver mutants to prove this hypothesis. These mutants could be used in SAXS to prove their hypothesis. Otherwise, I would remove the presented SAXS data.

3. The X-ray structure adds little value to the central tenant and should be moved to supplement Since this is consistent with prior structure, it can simply be stated that "in the absence of JM, pY1

forms a symmetric dimer, and occludes access of substrate". The overlay in supplemental 3a can then be presented in the main text.

4. Data for support of latch

I would move all of supplement figure 3 to the main text. This seems to be a main conclusion. However, given the fact that 407-420 binds the best and likely better than 414-430, the authors should include deletions of 407-414. Especially since the bulky Phe is at 410.

5. Interaction of CT with pY1

Fig 4c why not test binding of other CT truncations? Also, show differences here of KD and JM-KD with no mention. Isn't this a prelude to CT-JM interaction? State so here. Please present NMR data in a consistent manner throughout the manuscript. It should look identical in presentation to Fig 1e in style and format. Here the follow up binding experiments were thorough unlike the follow up to the NMR residues identified in the JM. On the contrary, given issues with precipitation and modest chemical shifts the authors should move Fig 5 to the supplement. Furthermore, NMR from fig5 did not form the basis of mutational follow up experiments for confirmation.

6. CT binding and impact on JM-KD

As opposed to JM binding studies, here SPR is employed. The data are conclusive that CT binds to pY1 with high affinity and slow (almost non existent) off rate. All other phospho forms fail to bind. The authors state that CT binds to unphospho KD Y657/658F, however the data show no binding. SPR can show an increase in background at high concentrations. The conclusion here, I assume would be the same for JM, that only pY1 shows significant binding. I would show the equilibrium SPR binding curve of CT to KD. This could be overlaid with all other non binding and moved to main figure. (instead of 6b SPR concentration series).

JM-CT direct binding. Again using MST there is poor basis for comparison. The authors conclude that 429-442 is best binding and therefore non-overlapping with JM's interaction with KD. However, the 407-420 shows highest percent shift response. Much greater than the 429-442. This begs the question, why do the authors not do SPR throughout for binding? Fits in supp 7b very poor and show kd greater than limit of assay at 1000000 nM.

Reviewer #2 (Remarks to the Author):

The authors present an extremely detailed study on the FGFR kinase and the involvement of regions adjacent to the kinase domains (the juxtamembrane region (JM) and C-terminal segment (CT)) in regulating its activity. The overall hypothesis of the authors is that these regions allow for an interaction between one FGFR molecule and another that adopts different orientations depending on the activation state of the receptor. To do this they have been unable to provide a direct structural snapshot of these different interaction schemes but rather have used an impressively wide array of biochemical techniques to gather the data needed for their final activation model.

Because of this, the manuscript is very difficult to follow. I understand that the experiments are complicated but even so I would suggest trying to re-write the results sections to make it easier to understand. There is an incredible amount of data in this paper. Because I found it so difficult to follow I've summarised what each figure shows in order to help me read it and to make sure I haven't missed anything:

Figure 1 shows:

- truncating the juxtamembrane region leads to a decrease in kinase activity in vitro and in cells
- increasing phosphorylation of KD leads to a decrease in the JM interaction (by pulldown and MST)
- regions of the N-lobe interact with JM as tracked by NMR

Figure 2 shows:

- increasing phosphorylation of KD leads to a decrease in dimer formation (by gel filtration, a lovely experiment!)
- truncating JM leads to an increase in dimerization as the kd (dimerization) is much lower in the absence of JM (by MST)

Figure 3 shows:

- the structure of monophosphorylated KD dimer and that it is similar to unphosphorylated KD (PDB 2PSQ)

Figure 4 shows:

- deletion of CT results in ligand-independent signalling and increased A-loop phosphorylation
- CT binds to KD and more tightly to JM-KD
- the proline rich region in the CT is likely responsible for binding

Figure 5 shows:

- residues in the KD involved in CT binding (by NMR), which do not completely overlap with JM-binding residues.

Figure 6 shows: (note there are 2 panel E's in Fig 6)

- CT blocks KD dimerization
- CT forms a high affinity complex with JM-KD
- CT interacts directly with JM

Figure 7 shows:

- the presence of JM attached to the active KD leads to enhanced activity whereas the presence of CT does the opposite
- as the JM-KD construct becomes more phosphorylated it interacts less with the CT-freeing it up for binding downstream effectors
- CT interferes with downstream FRS2 phosphorylation

Comments.

At times the certainty of their model is overstated and the data over-interpreted, especially the SAXS data. I would prefer language to be used that acknowledges the fact that none of the experiments "prove" that their working hypothesis is correct. Minor changes like "this data is consistent with a model in which....." Rather than "this shows that.....". The final model is very speculative.

I have a number of other comments/queries as listed below:

Results: opening paragraph reads more like something you would see in a thesis than a paper. But given the complexity of this work it is probably justified.

Affinity between different KDpY mutants and MBP-JM: Why do we need the raw data on Figure 1, the data as a bar graph in supp fig 1 and the same data as numbers in supp table 1? Also, please do not refer to Affinity Constant as having units of reciprocal micromolar – that is just confusing. Most people would consider the Kd to be the "affinity constant" and this is given in units of Molar (or micromolar if you prefer)-see Supp Fig 1c. There is no mention in Figure S1 as to what method was used to determine the affinities (MST).

Results: line 193: describing the peak shifts as "major" is over stating things. They shift barely 1 peakwidth. Maybe "noticeable" is a better word.

Results line 198 "CSPs <0.05 are indicative ofor fast on/off rates". No they are not. Titrations of weak-binding compounds (with fast on/off rates) is routine in NMR spectroscopy-usually, all that a low CSP values means is that you have not saturated the binding site.

Results, p9 line200-201. The sentence here is not really correct. Low values for CSPs are indicative of binding that has not saturated. The fact that your interaction is in fast-exchange (on the NMR timescale) is evidence of a fast off-rate, but that is not unusual. The fact that binding hasn't saturated is surprising given that you are many-fold above K_d in terms of protein concentration. But the K_d was only measured by MST and may have been underestimated. The data in SuppFig5 shows that even a 3-fold excess of JM is likely not saturating the kinase domain, which is presumably >300uM. There is a real discrepancy between the K_d as measured by MST and the K_d that NMR is suggesting.

Dimerisation of KDpY1-the K_d is quoted as being 112nM however the TROSY-HSQC spectrum is not consistent with a dimeric kinase domain (~70kDa) in my opinion. Did the authors ever run a HSQC on KD-JM? Is the linewidth different?

Results line 235: SAXS data infers a range of monomers, dimers, oligomers, with the "dimer corresponding to the symmetric dimer in PDB 2PSQ". This appears to be vastly overstating the certainty of this conclusion. There is no way that SAXS can tell you with certainty what the structure of the scattering species is-especially when it's a mixture of different oligomers.

Likewise, the presence of 20% symmetric dimers in the JM-KD structure can be only be inferred by SAXS, not directly shown.

Line 243: "The structures of the selected asymmetric dimers are reminiscent of....." they are reminiscent because you have chosen to use the asymmetric dimer model (PDB 3CLY) to fit your SAXS data.

Lines 250-254 is speculative and should be labelled as such.

Line 258: "these dimers correspond to the symmetric dimers observed in SAXS" this sentence should indicate that that is because you used PDB 2PSQ as a search model to fit the SAXS data.

The off-rate of the CT/JM-KD interaction is extremely slow. It would be very interesting to look at this by NMR as the CSPs would go into slow exchange-but I understand that is probably not feasible.

Reviewer #3 (Remarks to the Author):

The study aims to provide a comprehensive and intricate understanding of the steps of FGFR tyrosine kinase regulation focusing on the roles of the juxtamembrane and C-terminal tail regions in dimerization and activation. Unfortunately, the study overinterprets some of the data and seems to try to do too much, combined this makes it difficult to assess the impact of the proposed model(s). One wonders whether the authors may consider breaking this into separate studies that concisely and conclusively prove the distinct observations made. Some specific comments are listed below.

- 1) It is unclear what the phosphorylation state is for the FGFR constructs because statements regarding phosphorylation state are not quantitatively supported (the anti-phosphotyrosine blots not quantitative).
- 2) The SAXS seem to be over-interpreted. The SAXS lacks the requisite 'Table 1', Guinier plots, and other data quality analyses that are standard for such studies. No information is provided regarding observed MW, aggregation, or consistency of the data across concentrations. It seems unlikely that MultiFOXS analysis can deconvolute the conformational states as accurately as they state. They might

wish to conduct in-line SEC-MALS-SAXS to better understand this system.

3) The study attempts to make extensive and intricate claims regarding the regulation state of the FGFR kinase, the impact of JM and CT on specific phosphorylation states, the role of PxxP motifs, how adaptor binding impacts signaling, and the oligomerization states and mechanisms of transition between these states throughout the lifecycle of FGFR activation. In attempting to do so much the study fails to adequately prove many of its key insights, and should be split into a separate studies to demonstrate these observations conclusively.

Minor

- 1) The first two sentences of the abstract may not be perfectly accurate in all cases, and the authors may wish to modify to include a little wiggle-room.
- 2) For Figure 2 and Supplemental Figure 2b. Fig S2b should also show, on the same gel, the phosphorylation state of JM-KDK518I and JM-KDY657/658F.
- 3) MST data for Figure 1d and Supplementary Table 1 showing pY3 and pY5 are missing.
- 4) It is unclear what any of the stated errors refer to (SD/SEM, N).
- 5) The authors may want to consider the significant figures throughout – e.g. $13.4 \pm 0.994 \mu\text{M}$ in Table S1 would seem to be an over-statement of the accuracy of the measurements.
- 6) MST curve for the full JM 407-462 which yields $2.51 \pm 0.203 \mu\text{M}$ should be shown in Supp Fig 1c alongside the other 5 MST curves.
- 7) Figure legend for Figure 3 could be fleshed out. Fo-Fc maps should be included, and sigma level of the maps stated. In Supplementary Table 4, it might be helpful to also include Rpim. PDB validation report was not included in the submission.
- 8) Many of the figures are quite crowded with very small fonts.

Reviewers' Comments – Comms. Biol.

Reviewers' comments:

We should like to thank all three Reviewers for their efforts on this manuscript. In all cases the Reviewers have raised important and valuable points that we have done our best to address. We feel that thanks to their input we have improved the manuscript and made it of greater interest to the readership of Communications Biology. Based on the general view of the Reviewers' we have substantially reduced the Results section by >1000 words. We have also been able to reduce the number of Figures from seven to five and the number of Supplementary Figures also from seven to five.

Reviewer #1 (Remarks to the Author):

Lin et al in their manuscript "The combined action of the intracellular regions regulate oncogenic FGFR2 kinase activity" present an argument for the independent and combined regulatory actions of the JM (juxtamembrane) and CT (C-terminal) peptides upon the central enzymatically active KD (kinase domain) of the RTK (receptor tyrosine kinase) FGFR2 (fibroblast growth factor receptor II). Whilst their report does include some evidence in support of (rather consistent with) their central thesis, much presented data does not unequivocally support their conclusions, or the presented data is only tangentially related to their argument. This manuscript needs a substantial re write and re organization to make it more readable and to focus more on their main points.

We are very grateful to this Reviewer for their careful reading of the manuscript and their meticulous attention to the experimental detail. This Reviewer clearly has a good understanding the biophysical and structural techniques adopted in this work, and their comments have inspired us to critically analyse our data and ensure that these are presented correctly in the revised manuscript.

Main issues:

1. Data for sequential and progressive phosphorylation affecting JM binding to KD
The data presented from a pull down in 1c simply shows that pY1 is preferred to all other mutants. Why is there no WT here?

We apologise for the lack of clarity in this here. In this experiment we wanted to explore how the sequential phosphorylation of the phosphorylatable tyrosine residues (which are phosphorylated in the order 657, 587, 467, 589, 658, and 734) affected the ability of the kinase domain bind to JM. This is important because the JM has a regulatory role. The use of the intact wild type protein is prohibited because of solubility issues and this does not address the question re. binding of the JM . We have adopted the non-phosphorylated kinase represented by the kinase 'dead' form, i.e. the K518I mutant to represent the wild type interaction.

Why not show statistical significance?

See histogram below the blot which now shows the errors with respect to the monophosphorylated KD. We have also added the p-values to show the statistical significance.

Why Ponceau stain labeled MBP-JM? Shouldn't this be MBP-JM higher band, and MBP alone lower band?

We thank the Reviewer for spotting this and in accordance we have modified the labelling of this Figure to show MBP-JM is higher band and MBP is below.

Fig 1d seems to be the experiment that attempts to show quantitative differences between mutants. However, the fits are very poor. And the relative dF_{norm} for the different mutants is all different.

This Reviewer appears to have an understanding of the MST methodology, however some of the comments seem to show some minor misunderstanding of the data analysis and the appearance of typical fitting. Our equilibrium constants are derived from the best fits to the data and the errors are within the acceptable range for this technique in all cases.

Why doesn't each have same max change? They each have same mass and fold. More importantly, the fits are very poor. Apart from pY1 most can be fit to a straight line and do not approach a plateau at high concentration.

The MST technique is based on the movement of molecular species through a volume within a capillary tube. The maximum change will vary based on the molecular species and its environment. As a result the full scale deflections on the y-axis of the data will vary for each individual experiment. The fits we have obtained are good and very clearly are not straight lines. Our data fits are in line with previous reported data from MST (e.g. see <https://doi.org/10.1016/j.molstruc.2014.03.009>).

Together this makes me question the stability of all the other mutants. Why not do SPR for the binding assays. CT binding in Supp Fig 6b much more clear.

I would remove fig 1c and 1d from the main figs and place in supplement, or remove altogether.

We feel that both Figures 1c and 1d are important in demonstrating the impact of phosphorylation on the binding of JM to KD and inclusion of the two orthogonal approaches helps to underscore the validity of the data.

The data that is in supplement 1d is more convincing for binding. However, the fit is poor 414-430. In fact I am unsure how that was fit. It completely misses the high concentration point, and there is no indication of a plateau. My conclusion by looking at the data is that 407-420 binds with the highest affinity. And 414-430 is weaker by an order of magnitude.

The fitting determines the inflection at the concentration where the conditions for K_d are met. The K_d does not correlate with the ΔF_{norm} . The tighter binding peptide is 414-430.

However, additional controls need to be employed to show specificity. One worry with studies of small peptides is to ensure that it is not a non specific hydrophobic interaction. Here we see a direct pattern to binding with hydrophobicity. 407-420 as the only peptide with a bulky Phe. And 414-430 with 5 small hydrophobics. And the non binding peptides with substantially less hydrophobic character. What is necessary is a scramble peptide experiment of the binders to show specificity.

The Reviewer is right to point out that measuring the affinities of small peptide can be problematical. However, with respect to the point about binding of hydrophobic residues we agree that hydrophobic residues on the peptide can interact with similar residues on the KD. We have not discounted this in any way, and these interactions may be responsible for the differences seen in affinity between 407-420 and 414-430. The observation we draw from the data, i.e. that both peptides that show binding include the sequence PAVHKLT, does not preclude the Reviewer's point.

A scrambled peptide is unlikely to bind as seen with the other peptide sequences derived from other regions of the JM.

For figure 1e-f the authors use NMR to show interacting residues on the kinase domain. However the figures add no value for the reader. The whole KD is not shown and key residues identified by NMR are missing (N terminal prior to 468). Since nothing mechanistic can be gleaned from this, it can be moved to supplement.

We are grateful for the Reviewer's comments here (which are also in agreement with Reviewer #2). We agree that the attempt to map the binding of JM on KD is hampered by the small CSPs and difficulty in interpreting their source. As a result we have removed this experiment from the manuscript. This does not impinge on the overall findings of this study which is focused on the regions of JM and CT that are involved in binding, rather than where these might occur on KD. Knowledge of the binding site would be relevant if we were trying to map how the positioning of JM and CT binding on KD impacts substrate binding and subsequent kinase activity. This is beyond the scope of this study and would require optimising of the experimental protocol to improve the CSP data. We will likely pursue this in a future study.

And just stated as N-term residues up to 475 showed most shift. Were these NMR interactions tested with binding? Repeat experiment of supplement 1d with KD that lacks this 'critical' N-term. Or specific mutations of NMR identified residues.

Please see above.

2. SAXS data is not robust for showing the authors suggestions of different species
The dimerization constant presented for pY1 dimerization of 100nM (Fig 2c) means that at the concentrations used for SAXS there should be near complete dimerization. Showing residuals of plots to different imposed models does not confirm the symmetric or asymmetric orientations. The authors should generate bead models and dock, or rather use negative stain EM. If the authors really wanted to show asymmetric vs symmetric orientations they should use the classic experiments of donor receiver mutants to prove this

hypothesis. These mutants could be used in SAXS to prove their hypothesis. Otherwise, I would remove the presented SAXS data.

Based on this Reviewer's comments we have decided to remove the data derived from SAXS and publish it elsewhere.

3. The X-ray structure adds little value to the central tenant and should be moved to supplement

Since this is consistent with prior structure, it can simply be stated that "in the absence of JM, pY1 forms a symmetric dimer, and occludes access of substrate". The overlay in supplemental 3a can then be presented in the main text.

We have taken the Reviewer's advice and removed the X-ray structure to the Supplementary Materials. We have also followed the Reviewer's advice and used the statement provided in the revised text. We thank the Reviewer for this contribution to our manuscript.

4. Data for support of latch

I would move all of supplement figure 3 to the main text. This seems to be a main conclusion. However, given the fact that 407-420 binds the best and likely better than 414-430, the authors should include deletions of 407-414. Especially since the bulky Phe is at 410.

Again we have taken the Reviewer's advice and moved the Supplementary Figures to the main text. However, as we outline above we do not agree with the Reviewer's assessment of the binding of the JM peptides.

5. Interaction of CT with pY1

Fig 4c why not test binding of other CT truncations? Also, show differences here of KD and JM-KD with no mention. Isn't this a prelude to CT-JM interaction? State so here.

It is not clear what other CT truncations the Reviewer would suggest to be tested in Figure 4c. The two constructs used here simply demonstrate that CT binds to both isolated KD and JM-KD. At this point it would be confusing to intimate that JM and CT are binding before presenting further data that leads to this important conclusion. Nonetheless we are very gratified to see that the Reviewer has picked up on the potential for the JM to CT interaction.

Please present NMR data in a consistent manner throughout the manuscript. It should look identical in presentation to Fig 1e in style and format. Here the follow up binding experiments were thorough unlike the follow up to the NMR residues identified in the JM. On the contrary, given issues with precipitation and modest chemical shifts the authors should move Fig 5 to the supplement. Furthermore, NMR from fig5 did not form the basis of mutational follow up experiments for confirmation.

The NMR data in Figure 4 (now Figure 3) is presented in a different way to those in Figure 5 (now Figure 4) because it shows the CSPs for a titration of KD into CT. The different bars of the histogram correspond to different concentrations of KD. The other NMR data does not represent titration data hence it is portrayed differently.

6. CT binding and impact on JM-KD

As opposed to JM binding studies, here SPR is employed. The data are conclusive that CT binds to pY1 with high affinity and slow (almost non-existent) off rate. All other phospho forms fail to bind. The authors state that CT binds to unphospho KD Y657/658F, however the data show no binding. SPR can show an increase in background at high concentrations. The conclusion here, I assume would be the same for JM, that only pY1 shows significant binding. I would show the equilibrium SPR binding curve of CT to KD. This could be overlaid with all other non-binding and moved to main figure. (instead of 6b SPR concentration series).

We do show binding of CT to the non-phosphorylated KD Y657/658F. We agree that the binding measurement by SPR is conclusive in showing binding. We used equilibrium (rather than on/off rate) measurement, which could K_d could suffer from the possibility that we are not sure that equilibrium has been obtained under the concentration regime adopted in the experiment.

JM-CT direct binding. Again using MST there is poor basis for comparison. The authors conclude that 429-442 is best binding and therefore non-overlapping with JM's interaction with KD. However, the 407-420 shows highest percent shift response. Much greater than the 429-442. This begs the question, why do the authors not do SPR throughout for binding?

The affinity measurement is not based on the highest percentage shift response (as suggested by the Reviewer) but rather on the concentration of the inflection of the sigmoidal fit.

Fits in supp 7b very poor and show k_d greater than limit of assay at 1000000 nM.

The fitting for the data in Fig S7 is sufficient to show interactions are all below 1000000 nM (see inflection points of sigmoidal fits).

Reviewer #2 (Remarks to the Author):

The authors present an extremely detailed study on the FGFR kinase and the involvement of regions adjacent to the kinase domains (the juxtamembrane region (JM) and C-terminal segment (CT)) in regulating its activity. The overall hypothesis of the authors is that these regions allow for an interaction between one FGFR molecule and another that adopts different orientations depending on the activation state of the receptor. To do this they have been unable to provide a direct structural snapshot of these different interaction schemes but rather have used an impressively wide array of biochemical techniques to gather the data needed for their final activation model.

We are really grateful to this Reviewer for their careful reading of our manuscript and important observations. We appreciate that the Reviewer notes that we have performed "an extremely detailed study" and has fully understood the underlying hypothesis of this work.

Because of this, the manuscript is very difficult to follow. I understand that the experiments

are complicated but even so I would suggest trying to re-write the results sections to make it easier to understand. There is an incredible amount of data in this paper. Because I found it so difficult to follow I've summarised what each figure shows in order to help me read it and to make sure I haven't missed anything:

As the Reviewer has noted there are a lot of data in this work and we have had to use multiple techniques to confirm our observations. We wanted this manuscript to present an intact picture of the influence of both JM and CT on KD activity and self-association. Attempts to split this study to deal with JM and CT interactions independently left us without the opportunity to develop the final key finding that JM and CT interact with one another to provide an additional control mechanism.

In line with the Reviewer's comments we have modified the Results section to keep it succinct and removed some experimental data which is considered superfluous to the conclusions of the manuscript. This has shortened the text by approximately 1000 words and also reduced the number of Figures by two and Supplementary Figures by two.

Figure 1 shows:

- truncating the juxtamembrane region leads to a decrease in kinase activity in vitro and in cells
- increasing phosphorylation of KD leads to a decrease in the JM interaction (by pulldown and MST)
- regions of the N-lobe interact with JM as tracked by NMR

Correct!

Figure 2 shows:

- increasing phosphorylation of KD leads to a decrease in dimer formation (by gel filtration, a lovely experiment!) Thank you.
- truncating JM leads to an increase in dimerization as the kd (dimerization) is much lower in the absence of JM (by MST)

Correct!

Figure 3 shows:

- the structure of monophosphorylated KD dimer and that it is similar to unphosphorylated KD (PDB 2PSQ)

Correct!

Figure 4 shows:

- deletion of CT results in ligand-independent signalling and increased A-loop phosphorylation
- CT binds to KD and more tightly to JM-KD
- the proline rich region in the CT is likely responsible for binding

Correct!

Figure 5 shows:

- residues in the KD involved in CT binding (by NMR), which do not completely overlap with JM-binding residues.

Correct!

Figure 6 shows: (note there are 2 panel E's in Fig 6)

- CT blocks KD dimerization
- CT forms a high affinity complex with JM-KD
- CT interacts directly with JM

Correct!

Figure 7 shows:

- the presence of JM attached to the active KD leads to enhanced activity whereas the presence of CT does the opposite
- as the JM-KD construct becomes more phosphorylated it interacts less with the CT-freeing it up for binding downstream effectors
- CT interferes with downstream FRS2 phosphorylation

Correct!

The Reviewer has carefully read the manuscript and completely understands the many key experiments and their outcome that go together to enable us to make the important conclusions from this work.

Comments.

At times the certainty of their model is overstated and the data over-interpreted, especially the SAXS data. I would prefer language to be used that acknowledges the fact that none of the experiments “prove” that their working hypothesis is correct. Minor changes like “this data is consistent with a model in which.....” Rather than “this shows that.....”. The final model is very speculative.

We have understood the Reviewer's comments and made changes accordingly. This has included removing the SAXS data in its entirety and changing the wording to be compliant with the Reviewer's concerns about proving our hypothesis. We feel, however, that the final model is consistent with the data which are gathered on the interactions we have characterised and provide a useful insight and future hypothesis to be tested. We also feel that it is consistent with recent published work on the CT regulation of FGFR2 in animal models (Nature. 2022 Aug;608(7923):609-617. doi: 10.1038/s41586-022-05066-5.)

I have a number of other comments/queries as listed below:

Results: opening paragraph reads more like something you would see in a thesis than a paper. But given the complexity of this work it is probably justified.

We have shortened this paragraph, however we feel that an overview of the approach is important because of the size and complexity of this study.

Affinity between different KDpY mutants and MBP-JM: Why do we need the raw data on Figure 1, the data as a bar graph in supp fig 1 and the same data as numbers in supp table 1? Also, please do not refer to Affinity Constant as having units of reciprocal micromolar – that is just confusing. Most people would consider the Kd to be the “affinity constant” and this is given in units of Molar (or micromolar if you prefer)-see Supp Fig 1c. There is no mention in Figure S1 as to what method was used to determine the affinities (MST).

We apologise for any confusion here. We have rectified this problem

We presented the MST curves in the main Fig. 1. and the reviewer is correct that the bar graph in Supp Fig. 1 reflects the same data as in Supp Table 1. We used the bar graph to with a view to clarifying the data. We apologise for any confusion and we have now removed this graph to keep our message simple and clear.

Results: line 193: describing the peak shifts as “major” is over stating things. They shift barely 1 peakwidth. Maybe “noticeable” is a better word.

We have changed the wording as suggested by the Reviewer.

Results line 198” “CSPs <0.05 are indicative ofor fast on/off rates”. No they are not. Titrations of weak-binding compounds (with fast on/off rates) is routine in NMR spectroscopy-usually, all that a low CSP values means is that you have not saturated the binding site.

We have removed this sentence in compliance with the Reviewer’s comment.

Results, p9 line200-201. The sentence here is not really correct. Low values for CSPs are indicative of binding that has not saturated. The fact that your interaction is in fast-exchange (on the NMR timescale) is evidence of a fast off-rate, but that is not unusual. The fact that binding hasn’t saturated is surprising given that you are many-fold above K_d in terms of protein concentration. But the K_d was only measured by MST and may have been underestimated. The data in SuppFig5 shows that even a 3-fold excess of JM is likely not saturating the kinase domain, which is presumably >300uM. There is a real discrepancy between the K_d as measured by MST and the K_d that NMR is suggesting.

The Reviewer makes a very good point here, and based on these comments we have removed the experiments on the JM binding to KD. This does not impinge on the overall findings of this study which is focused on the regions of JM and CT that are involved in binding, rather than where these might occur on KD. Knowledge of the binding site would be relevant if we were trying to map how the positioning of JM and CT binding on KD impacts substrate binding and subsequent kinase activity. This is beyond the scope of this study and would require optimising of the experimental protocol to improve the CSP data. We will likely pursue this in a future study.

Dimerisation of KDpY1-the K_d is quoted as being 112nM however the TROSY-HSQC spectrum is not consistent with a dimeric kinase domain (~70kDa) in my opinion. Did the authors ever run a HSQC on KD-JM? Is the linewidth different?

Results line 235: SAXS data infers a range of monomers, dimers, oligomers, with the “dimer corresponding to the symmetric dimer in PDB 2PSQ”. This appears to be vastly overstating the certainty of this conclusion. There is no way that SAXS can tell you with certainty what the structure of the scattering species is-especially when it’s a mixture of different oligomers.

Because of the apparent conjecture regarding our SAXS data we have removed these results for publication elsewhere. Removing these data do not affect the conclusions of this section on JM-KD dimerization, or of the overall study.

Likewise, the presence of 20% symmetric dimers in the JM-KD structure can be only be inferred by SAXS, not directly shown.

See above.

Line 243: “The structures of the selected asymmetric dimers are reminiscent of.....” they are reminiscent because you have chosen to use the asymmetric dimer model (PDB 3CLY) to fit your SAXS data.

See above.

Lines 250-254 is speculative and should be labelled as such.

This sentence has been modified to reflect that we speculate on the function of the latch in allowing KDs to orientate with respect to one another.

Line 258: “these dimers correspond to the symmetric dimers observed in SAXS” this sentence should indicate that that is because you used PDB 2PSQ as a search model to fit the SAXS data.

This has been removed and the crystal structure has been reported as Supplementary Material.

The off-rate of the CT/JM-KD interaction is extremely slow. It would be very interesting to look at this by NMR as the CSPs would go into slow exchange-but I understand that is probably not feasible.

We have considered this, but as the Reviewer points out the experiment is not easy to interpret.

Reviewer #3 (Remarks to the Author):

The study aims to provide a comprehensive and intricate understanding of the steps of FGFR tyrosine kinase regulation focusing on the roles of the juxtamembrane and C-terminal tail regions in dimerization and activation. Unfortunately, the study overinterprets some of the data and seems to try to do too much, combined this makes it difficult to assess the impact of the proposed model(s). One wonders whether the authors may consider breaking this into separate studies that concisely and conclusively prove the distinct observations made. Some specific comments are listed below.

We thank this Reviewer for their careful reading of the manuscript. We have made a major effort to ensure that our study does not over-interpret the data. We have also taken out some of the experiments where we felt that these might provide data that does not add to the overall conclusions of the study. For example, we have removed the SAXS experiments and moved the X-ray structural detail to the Supplementary Materials.

We did attempt to re-write the study as two separate manuscripts, however dealing with JM and CT interactions independently left us without the opportunity to develop the final key model that JM and CT interact with one another to provide an additional control mechanism.

1) It is unclear what the phosphorylation state is for the FGFR constructs because statements regarding phosphorylation state are not quantitatively supported (the anti-phosphotyrosine blots not quantitative).

We thank the Reviewer for this comment. We have previously established the phosphorylation protocol in our lab and use mass spectrometry to quantify the phosphorylation states of recombinant FGFR2 proteins. However, we found that it is impossible to get uniformed, 100% phosphorylated (or dephosphorylated) proteins. This could be due to the low phosphorylation stoichiometry of certain tyrosines and the instability of the phosphodiester bond in some phosphotyrosines. To overcome this we therefore adapted the method to create clones according to the phosphorylation order of FGFRs which allow us to phosphorylate FGFR2 proteins at the maximum level at each stage of the sequential phosphorylation.

2) The SAXS seem to be over-interpreted. The SAXS lacks the requisite 'Table 1', Guinier plots, and other data quality analyses that are standard for such studies. No information is provided regarding observed MW, aggregation, or consistency of the data across concentrations. It seems unlikely that MultiFOXS analysis can deconvolute the conformational states as accurately as they state. They might wish to conduct in-line SEC-MALS-SAXS to better understand this system.

We have removed the SAXS data from the manuscript for publication elsewhere.

3) The study attempts to make extensive and intricate claims regarding the regulation state of the FGFR kinase, the impact of JM and CT on specific phosphorylation states, the role of PxxP motifs, how adaptor binding impacts signaling, and the oligomerization states and mechanisms of transition between these states throughout the lifecycle of FGFR activation. In attempting to do so much the study fails to adequately prove many of its key insights, and should be split into a separate studies to demonstrate these observations conclusively.

See above regarding the separation of the work into more than one manuscript.

Minor

1) The first two sentences of the abstract may not be perfectly accurate in all cases, and the authors may wish to modify to include a little wiggle-room.

We have modified the opening sentence of the abstract.

2) For Figure 2 and Supplemental Figure 2b. Fig S2b should also show, on the same gel, the phosphorylation state of JM-KDK518I and JM-KDY657/658F.

We have shown the phosphorylation state of KD^{K518I} in the supplementary figure 1. As K518I and Y657/658F are both well-established kinase-dead mutants and have been used widely by us and other researchers we, therefore, did not include them in Supplementary Fig. 2.

3) MST data for Figure 1d and Supplementary Table 1 showing pY3 and pY5 are missing. We did not conduct MST for pY3 and pY5 as the pulldown already showed that these two mutants bind to JM with lower affinity than the pY1 which is the focus here.

4) It is unclear what any of the stated errors refer to (SD/SEM, N). They refer to standard deviation (SD). We have added the information to the Methods section.

5) The authors may want to consider the significant figures throughout – e.g. $13.4 \pm 0.994 \mu\text{M}$ in Table S1 would seem to be an over-statement of the accuracy of the measurements

We have complied with this comment.

6) MST curve for the full JM 407-462 which yields $2.51 \pm 0.203 \mu\text{M}$ should be shown in Supp Fig 1c alongside the other 5 MST curves.

Although we appreciate the sentiment of this comment we have decided to keep the full JM 407-462 ($2.51 \pm 0.203 \mu\text{M}$, titrated with KD^{pY1}) curve in the main figure with other full-length JM – KD (different pY states). We kept the 5 MST curves in the Supplementary Fig 1 because these are truncated forms and we feel that this will avoid confusion because these are two distinct groups of different JM peptides.

7) Figure legend for Figure 3 could be fleshed out. Fo-Fc maps should be included, and sigma level of the maps stated. In Supplementary Table 4, it might be helpful to also include Rpim. PDB validation report was not included in the submission.

We have removed this Figure and the structural detail.

8) Many of the figures are quite crowded with very small fonts.

We have reduced the numbers of figures from seven to five and have enlarged figures where appropriate to include clearer fonts.

Reviewers' comments:

Reviewer #1 (Remarks to the Author):

The study attempts to make a series of detailed mechanistic claims about how the JM and CT domains affect the activity of the KD (and its different phos states). The authors make leaps from their data about how the structure may be regulating the KD. It would be best to split these into separate stories and to do a more careful analysis of each proposed mechanism. In the current state it is too difficult to derive any coherent story and the data do not strongly support many of the claims. The authors need to do more careful binding analysis (many fits are poor and still in linear range). Scrambled peptides must be employed to rule out non specific hydrophobic interactions.

Some specific points (but not exhaustive): Figures should show all data and not selected data: Fig 1d missing pY5. Why no py3? SuppFig1 407-420 seems the best binder. The Kd of 414-430 is not fit well. Kd is half maximal of Fnorm, but authors did not reach a maximum. This is regardless if the ligand induces a change in initial fluorescence, which could shift Kd from inflection point. But the Max still needs to be determined. Fig 2 should have KDpy1 on same plot as Fig 2a. All should be done properly in same way. As is, JM KDpy1 shows mostly dimer in 2a and mostly monomer in 2b. Fig2a inset colors not same as in trace. The fact that KDpy1 has an apparent Kd of 112nM means that all would expected to be dimer in SEC, but it is not. The western in 2d and 2e seem to show all active kinases regardless of JM. No repeats or significance shown. This is a small change. Fig3a no significance shown, effect seems minor. Fig 3c is actually py1 but not labeled as such. And in text there is only 1 line describing Fig3c and it avoids the fact that JM-KDpy1 binds much better than KDpy1. Only to come back to this many pages later.

Reviewer #2 (Remarks to the Author):

The authors have made significant changes to the manuscript which makes it far easier to read and interpret and all of my comments/queries have been satisfactorily addressed. Congratulations to the authors for an extremely comprehensive body of work.

Also, I'd like to take this opportunity to thank John Ladbury (to be clear to the editor we we have never met or interacted in any way) for decades of excellent biochemical/biophysical research-his pioneering papers on SH2 domains has been extremely valuable for my research over the past 20 years.

Reviewer #3 (Remarks to the Author):

The authors have been responsive to my comments and have appropriately modified the manuscript. I have no further concerns.

Reviewer #1:

The study attempts to make a series of detailed mechanistic claims about how the JM and CT domains affect the activity of the KD (and its different phos states). The authors make leaps from their data about how the structure may be regulating the KD.

We are very grateful to this Reviewer for their comments and continued efforts to improve this manuscript.

It would be best to split these into separate stories and to do a more careful analysis of each proposed mechanism.

We did attempt to re-write the study as two separate manuscripts, however dealing with JM and CT interactions independently left us without the opportunity to develop the final key model that JM and CT interact with one another to provide an additional control mechanism.

In the current state it is too difficult to derive any coherent story and the data do not strongly support many of the claims.

In response to the Reviewer's previous comments, we have dramatically revised and shortened the manuscript. We have substantially reduced the Results section by >1000 words. We have also been able to reduce the number of Figures from seven to five and the number of Supplementary Figures also from seven to five. Furthermore, we complied with this Reviewer's comments re. our X-ray structural data and removed this from the manuscript for publication elsewhere. These changes, we feel, have made the manuscript more coherent.

The authors need to do more careful binding analysis (many fits are poor and still in linear range). Scrambled peptides must be employed to rule out non specific hydrophobic interactions.

We thank the Reviewer for these comments; however, it is not clear what particular data he/she is referring to. In their previous review, this Reviewer highlighted the MST data as not being well fit, however we strongly disagree with this conclusion and highlighted previous published data to exemplify the expected fitting for this technique. It should be emphasised that the binding that we are measuring is generally weak to moderate affinity. The fits, as described by the inflection points, are clear to see. The reported data in the Supplementary Tables includes the fitting errors, so that the reader can judge the quality of the fit and the accuracy of the reported affinities.

Taking each dataset in turn the following comments are pertinent:

Figure 1d – These data reveal that increasing the number of phosphorylation sites on the KD reduces the affinity for binding to JM. Looking at the inflections of these curves, the numbers on the x-axis show a clear right-shift from pY1 to the higher phosphorylated states (i.e., pY2 – pY6). The higher phosphorylated states all show weak binding by greater than an

order of magnitude. Thus, we are able to confidently say that the affinity of the monophosphorylated state is tighter than the higher phosphorylated KD forms. This observation is very clearly backed up by the pull-down assay shown in **Figure 1c**. Based on these data we feel justified in making the statement in the text “Only weak binding is apparent with a non-phosphorylatable catalytically inactive K518I mutant, KD^{K518I}. The affinity of JM for KD reduces with progressive phosphorylation.”

Figure 2c – These data report on the apparent dissociation constants of the monophosphorylated KD and the monophosphorylated JM-KD. Hopefully the Reviewer has no problem with the quality of these fits. Assuming this is the case, then we feel justified in making the statement in the manuscript: “Thus, the presence of JM reduces the dimerization affinity by an order of magnitude, suggesting that KD^{pY1} and JM-KD^{pY1} dimers are conformationally different, and that the presence of JM reduces the ability of KD to tightly self-associate.”

Figure 3c – These data report on the binding of CT to KD and JM-KD. Again, hopefully the Reviewer is happy that these data cannot be fit by straight lines and therefore, we are confident in our comment “CT^{C1} binds to KD^{pY1} with moderate affinity ($K_d = 3.75 \pm 0.46 \mu\text{M}$)”.

Figure 4g – This MST experiment shows the direct binding between JM and CT. Again, we hope that the Reviewer will agree that this is a suitable fit and not a straight line. This direct binding is validated by the NMR data in **Figure 4h**.

Supplementary Figure 1b – These data report on the binding of short peptides derived from the JM to KD. Two out of the five peptides show evidence for binding. We have modified the text to comply with the Reviewer’s comments.

i.e.,

“The tightest binding peptide binds approximately an order of magnitude weaker than the intact JM and contained the sequence ⁴¹⁴PAVHKLTKRIPRRQVT⁴³⁰, $K_d = 36.8 \pm 6.2 \mu\text{M}$. The sequence ⁴⁰⁷KPDFSSQPAVHKLT⁴²⁰ bound approximately three-fold weaker; binding to the other sequences was negligible.”

Has been changed to

“Only two peptides bound to KD^{pY1}, (Supplementary Table S2). The two binding peptides share the consensus sequence ⁴¹⁴PAVHKLT⁴²⁰ which is proximal to the N-terminal of JM.”

Supplementary Figure 3a – These data report the binding of short peptides derived from the last 23 residues of CT binding to monophosphorylated KD. These are aimed at identifying the precise sequence that binds to KD^{pY1}. These are weak interactions (i.e. > 25 μM). The Reviewer should agree that the fit for the sequence CT⁸⁰¹⁻⁸¹⁵ is not a straight line. Thus, we are able to confidently make the comment in the text “MST affinity measurements of peptide fragments of CT revealed that the tightest binding sequence was ⁸⁰¹PDPMPYEPCLPQYPH⁸¹⁵”.

Supplementary Figure 5a – These data identify the JM sequence that binds to CT. Short peptides are derived from the JM. The fits of four out of the five plots are clearly not straight lines. These binding events allow us to confidently make the comment in the text “We also identified that the highest affinity sequence of JM that recognized CT includes residues ⁴²⁹VTVSAESSSSMNSN⁴⁴²”

Supplementary Figure 5b – These experiments report on the binding of CT to JM with a view to identifying the CT sequence involved. Two MST experiments show evidence of binding. These both have the common sequence of residues 808-815. Therefore, we are able to state “Using a series of short peptides derived from CT we demonstrated that the proline-rich sequence from CT binds to JM (Supplementary Table S7 and Supplementary Fig. 5b), and ⁸⁰⁸PCLPQYPH⁸¹⁵ sequence is necessary for CT to bind to JM”.

Some specific points (but not exhaustive): Figures should show all data and not selected data: Fig 1d missing pY5. Why no py3?

The data shown in **Figure 1d** is not selected. These data (as stated above) are included to show that there is a general weakening of the interaction between JM and KD as the KD is increasingly phosphorylated. The Figure clearly shows a clear right-shift from pY1 to the higher phosphorylated states (i.e. pY2, pY4, pY6). We did not make the measurement for pY3 or pY5 because we felt that it is appropriate to conclude that these will also show weaker binding than pY1. This presumption is clearly supported by the pull-down data in **Figure 1c** which includes pY3 and pY5.

SuppFig1 407-420 seems the best binder. The Kd of 414-430 is not fit well. Kd is half maximal of Fnorm, but authors did not reach a maximum. This is regardless if the ligand induces a change in initial fluorescence, which could shift Kd from inflection point. But the Max still needs to be determined.

We have commented on the fitting above. Based on the Reviewer’s comments we have modified the text to remove comment on the comparison of the two peptides.

It is not clear what the Reviewer means by “the Max still needs to be determined”. The MST technique is based on the movement of molecular species through a volume within a capillary tube. The maximum change will vary based on the molecular species and its environment. As a result, the full-scale deflections on the y-axis of the data will vary for each individual experiment. The fits we have obtained are good and very clearly are not straight lines. Our data fits are in line with previous reported data from MST.

Fig 2 should have KDpy1 on same plot as Fig 2a. All should be done properly in same way. As is, JMKDpy1 shows mostly dimer in 2a and mostly monomer in 2b. Fig2a inset colors not same as in trace. The fact that KDpy1 has an apparent Kd of 112nM means that all would be expected to be dimer in SEC, but it is not.

The two plots shown in Figures 2a and 2b are done with different columns and under different concentration regimes, hence they are not superimposable.

Thanking the Reviewer for spotting this, we have changed the colours in the inset to align with the plots.

The reason that we do not see all of the construct as dimer in the SEC is that the concentration on the column is initially 1-5 μM (i.e., close to the K_d).

The western in 2d and 2e seem to show all active kinases regardless of JM. No repeats or significance shown. This is a small change.

In both experiments an active kinase is included to enable us to see the effects of JM on kinase activity. This clearly shows that the presence of the JM on the active kinase increases turnover of phosphorylation on the inactive kinase substrate. This concurs with the conclusion derived from EGFR (see Ref 2).

Fig3a no significance shown, effect seems minor.

We thank the Reviewer for this comment. We have added the statistical p-value to show significance of the data.

Fig 3c is actually py1 but not labeled as such.

We have corrected the label as requested.

And in text there is only 1 line describing Fig3c and it avoids the fact that JM-KDpy1 binds much better than KDpy1. Only to come back to this many pages later.

We apologise if this has caused the Reviewer confusion here, however we wanted to avoid including two separate MST figures. We will modify this at the Editors discretion.

Reviewer #2 (Remarks to the Author):

The authors have made significant changes to the manuscript which makes it far easier to read and interpret and all of my comments/queries have been satisfactorily addressed. Congratulations to the authors for an extremely comprehensive body of work.

We are very grateful to this Reviewer for effort on this manuscript and for their comments regarding this work.

Also, I'd like to take this opportunity to thank John Ladbury (to be clear to the editor we we have never met or interacted in any way) for decades of excellent biochemical/biophysical research-his pioneering papers on SH2 domains has been extremely valuable for my research over the past 20 years.

Wow! I have never seen this before in a Review. I am very grateful for this comment. It is extremely gratifying to be commended in this way by a Reviewer.

Reviewer #3 (Remarks to the Author):

The authors have been responsive to my comments and have appropriately modified the manuscript. I have no further concerns.

We are grateful to this Reviewer for their efforts on our behalf and for their final comments.